# Online Preference Alignment for Language Models via Count-based Exploration

**Chenjia Bai**[1,6], **Yang Zhang**[2,1], **Shuang Qiu**[3], **Qiaosheng Zhang**[4], **Kang Xu**[5], **Xuelong Li**[1]*

[1]Institute of Artificial Intelligence (TeleAI), China Telecom, [2]Tsinghua University,
[3]City University of Hong Kong, [4]Shanghai AI Laboratory, [5]Tencent AI Lab
[6]Shenzhen Research Institute of Northwestern Polytechnical University
 baicj@chinatelecom.cn, xuelong_li@ieee.org

## Abstract

Reinforcement Learning from Human Feedback (RLHF) has shown great potential in fine-tuning Large Language Models (LLMs) to align with human preferences. Existing methods perform preference alignment from a fixed dataset, which can be limited in data coverage, and the resulting reward model is hard to generalize in out-of-distribution responses. Thus, online RLHF is more desirable to empower the LLM to explore outside the support of the initial dataset by iteratively collecting the prompt-response pairs. In this paper, we study the fundamental problem in online RLHF, i.e. *how to explore* for LLM. We give a theoretical motivation in linear reward assumption to show that an optimistic reward with an upper confidence bound (UCB) term leads to a provably efficient RLHF policy. Then, we reformulate our objective to direct preference optimization with an exploration term, where the UCB-term can be converted to a count-based exploration bonus. We further propose a practical algorithm, named *Count-based Online Preference Optimization (COPO)*, which leverages a simple coin-flip counting module to estimate the pseudo-count of a prompt-response pair in previously collected data. COPO encourages LLMs to balance exploration and preference optimization in an iterative manner, which enlarges the exploration space and the entire data coverage of iterative LLM policies. We conduct online RLHF experiments on Zephyr and Llama-3 models. The results on instruction-following and standard academic benchmarks show that COPO significantly increases performance.

## 1 Introduction

Reinforcement Learning from Human Feedback (RLHF) is a key tool to align the behaviors of Large Language Models (LLMs) with human values and intentions (Christiano et al., 2017; Bai et al., 2022b; Ouyang et al., 2022). By fine-tuning the pre-trained LLMs using human-labeled preference data, RLHF achieves enhanced performance and robustness to ensure that they operate consistently with user expectations. Existing RLHF methods focus mainly on preference alignment in an offline dataset by estimating reward functions through the Bradley-Terry (BT) model (Bradley & Terry, 1952) and performing policy gradients to update the LLM as a policy (Ahmadian et al., 2024) to maximize rewards. Other methods perform Direct Preference Optimization (DPO) (Rafailov et al., 2023; Meng et al., 2024) that considers a problem of restricted reward maximization to define the preference loss in LLM policy. However, both types of methods perform offline preference alignment in a fixed dataset, which can be limited in the coverage of the preference data. As discussed in theoretical work (Xiong et al., 2024), learning an optimal policy through offline RLHF requires a preference dataset with uniform coverage over the entire prompt-response space, which is impossible to satisfy for existing preference datasets. Thus, the offline learned explicit or implicit reward model cannot accurately estimate the reward of prompt-response pairs that are out-of-distribution (OOD) concerning the dataset, limiting the capacities of aligned LLMs (Rafailov et al., 2024a).

To address this problem, recent work attempts to extend offline RLHF to an online RLHF process by (i) generating new responses using the current LLM policy, (ii) obtaining preference labels from

---
*Corresponding Author

human or AI feedback, and (iii) performing preference alignment to update the LLM. The above steps can be repeated for several iterations to enhance the capabilities of LLMs. We highlight that the central problem in an online RLHF process is *how to explore* the prompt-response space in each iteration. Considering an extreme case where the LLM is a deterministic policy without exploration ability, the preference data collected in a new iteration will not increase the coverage of preference data, which means that the LLM policy cannot be improved via multiple iterations. Similar to the exploration problem in the standard online Reinforcement Learning (RL) problem (Kearns & Singh, 2002; Houthooft et al., 2016), systematic exploration in online RLHF is also important to efficiently explore the large space of token sequences and to collect informative experiences that could benefit policy learning the most. Recently, several works have tried to address this problem by introducing an optimism term in reward or value estimation (Dwaracherla et al., 2024; Cen et al., 2024), guide the policy towards potentially high-reward responses (Zhang et al., 2024), or actively explore out-of-distribution regions (Xie et al., 2024). However, they mostly rely on the likelihood derived by the LLM itself to encourage the policy to be different from the policies in previous iterations, which lacks theoretical guarantees to empower the LLM to explore systematically based on the confidence of the learned reward model for prompt-response pairs.

In this paper, we propose a novel algorithm, named *Count-based Online Preference Optimization (COPO)*, for efficient exploration in online RLHF. Specifically, COPO extends count-based exploration (Strehl & Littman, 2008; Bellemare et al., 2016) that is provably efficient in online RL to online RLHF for systematic exploration of LLM. We start by constructing an optimistic RLHF problem with an optimistic reward function in a confidence set of the reward. In the linear reward assumption, our result shows that the reward with an upper-confidence bound (UCB) bonus leads to a provably efficient RLHF policy with $\tilde{O}(\sqrt{T})$-regret bound. Then, we covert the UCB term to a general pseudo-count of prompt-response pair under the tabular MDP settings, which serve as a special case of the linear case and make the UCB term easy to estimate. Finally, we formulate the optimistic objective that is theoretically grounded under DPO reward parameterization in general cases, which results in an optimization objective that combines the DPO objective and count-based exploration, where we adopt a differentiable coin-flip counting network (Lobel et al., 2023) to estimate the pseudo-counts of prompt-response pairs via simple supervised regression.

Our contributions can be summarized as follows: (i) We propose COPO to encourage the LLMs to balance exploration and preference optimization in an iterative manner, which enlarges the exploration space and whole data coverage of the iterative LLM policies; (ii) We construct a lightweight pseudo-counting module with several fully-connected layers based on the LLM, which is theoretically grounded in policy optimization of online RLHF; (iii) we conduct RLHF experiments of COPO and several strong online RLHF baselines on Zephyr and Llama-3 models. The results of instruction-following and academic benchmarks show that COPO achieves better performance.

## 2 Preliminaries

We present the standard RLHF pipeline, summarized from the standard LLM alignment workflow. Specifically, a language model takes a prompt denoted by $x \in \mathcal{X}$, and generates a response denoted $y \in \mathcal{Y}$. Accordingly, we can take $\mathcal{X}$ as the state space of the contextual bandit and $\mathcal{Y}$ as the action space, and consider the language model as a policy $\pi$ which maps $x$ to a distribution over $\mathcal{Y}$. The standard RLHF process typically comprises 3 stages built on a reference LLM $\pi_{\text{ref}}$: (i) collecting preference data with the aid of a human labeler or scoring model, (ii) modeling the reward function from the preference data, and (iii) fine-tuning the LLM initialized from $\pi_{\text{ref}}$ via RL.

**Reward Modeling from Preference Data.** Following Ouyang et al. (2022); Rafailov et al. (2023); Zhu et al. (2023), we assume that there exists a ground-truth reward function $r^*(x, y) : \mathcal{X} \times \mathcal{Y} \to \mathbb{R}$ and the preference induced by the reward function satisfies the BT model, as

$$\mathbb{P}(y_1 \succ y_2 | x) = \frac{\exp(r^*(x, y_1))}{\exp(r^*(x, y_1)) + \exp(r^*(x, y_2))} = \sigma\left(r^*(x, y_1) - r^*(x, y_2)\right),$$

where $y_1 \succ y_2$ means $y_1$ is preferred over $y_2$, and $\sigma(z) := 1/(1 + \exp(-z))$ is the sigmoid function. Hence, a preference pair can be denoted by a tuple $(x, y_w, y_l)$, where $y_w$ and $y_l$ denotes the preferred and dispreferred response amongst $(y_1, y_2)$ respectively.

Given a preference dataset $\mathcal{D} = \{(x, y_w, y_l)\}$ sampled from $\mathbb{P}$, we can estimate the reward function $r$ via maximum likelihood estimation,

$$r_{\text{MLE}} = \arg\max_r \sum_{(x, y_w, y_l) \in \mathcal{D}} \log \sigma\big(r(x, y_w) - r(x, y_l)\big). \tag{1}$$

**RL Fine-tuning.** In this stage, we fine-tune the language model with the feedback provided by the reward function $r$. In particular, the goal of the language model $\pi$ is to maximize the reward while remaining close to the initial reference language model $\pi_{\text{ref}}$, thereby formulating the KL-regularized optimization problem which maximizes the following objective,

$$J(\pi, r) = \mathbb{E}_{x \sim \rho, y \sim \pi(\cdot|x)}\left[r(x, y)\right] - \beta \mathbb{E}_{x \sim \rho}\big[D_{\text{KL}}(\pi(\cdot|x)\|\pi_{\text{ref}}(\cdot|x))\big], \tag{2}$$

where $D_{\text{KL}}(\pi(\cdot|x)\|\pi_{\text{ref}}(\cdot|x))$ is the Kullback-Leibler (KL) divergence from $\pi$ to $\pi_{\text{ref}}$, $\beta > 0$ is the KL penalty coefficient, and $\rho$ is the distribution prompt $x$ sampled from. As a result, the RL fine-tuned LLM $\pi_r$ w.r.t. a given reward function $r$ is computed via,

$$\pi_r := \arg\max_\pi J(\pi, r) = \arg\max_\pi \mathbb{E}_{x \sim \rho, y \sim \pi(\cdot|x)}\left[r(x, y) - \beta \log \frac{\pi(y|x)}{\pi_{\text{ref}}(y|x)}\right].$$

Due to the discrete nature of language generation, this objective is not differentiable and is typically optimized with RL algorithms. The classic approaches (Ziegler et al., 2019; Ouyang et al., 2022; Bai et al., 2022b) construct the reward $\hat{r}(x, y) = r(x, y) - \beta(\log \pi(y|x) - \log \pi_{\text{ref}}(y|x))$, and maximize it using Proximal Policy Optimization (PPO) (Schulman et al., 2017).

**Direct Preference Optimization.** Alternatively, the KL-regularized objective in Eq. (2) admits a closed-form solution as

$$\pi_r(y|x) = \frac{\pi_{\text{ref}}(y|x) \exp(r(x, y)/\beta)}{Z(r, x)}, \tag{3}$$

where $Z(r, x) = \sum_y \pi_{\text{ref}}(y|x) \exp(r(x, y)/\beta)$ is the partition function. Eq. (3) in turn reparameterizes the reward function $r$ as,

$$r(x, y) = \beta\big(\log \pi_r(y|x) - \log \pi_{\text{ref}}(y|x) + \log Z(r, x)\big). \tag{4}$$

Motivated by this reparameterization, DPO (Rafailov et al., 2023) substitutes Eq. (4) into the reward MLE (Eq. (1)), and integrates reward modeling and RL fine-tuning into a single policy optimization objective. DPO bypasses the need for explicitly learning the reward and optimization objective is

$$\arg\min_{\pi_\theta} \mathcal{L}_{\text{DPO}}(\pi_\theta; \mathcal{D}) = \arg\min_{\pi_\theta} - \sum_{(x, y_w, y_l) \in \mathcal{D}} \log \sigma\left(\beta \log \frac{\pi_\theta(y_w|x)}{\pi_{\text{ref}}(y_w|x)} - \beta \log \frac{\pi_\theta(y_l|x)}{\pi_{\text{ref}}(y_l|x)}\right). \tag{5}$$

## 3 COUNT-BASED ONLINE PREFERENCE OPTIMIZATION

### 3.1 THEORETICAL MOTIVATION

We use calligraphic letters for sets, e.g., $\mathcal{S}$ and $\mathcal{A}$. Given a set $\mathcal{S}$, we write $|\mathcal{S}|$ to represent the cardinality of $\mathcal{S}$. For vectors $a$ and $b$, we use $\langle a, b \rangle = a^\top b$ to denote their inner product. We write $\|a\|_\Sigma = \sqrt{a^\top \Sigma a}$ as a semi-norm of $a$ when $\Sigma$ is some positive semi-definite matrix. We focus on linear reward settings for our theoretical motivation, and we will provide a practical algorithm in the next section. Formally, we make the following assumption on the parameterization of the reward.

**Assumption 1** (Linear Reward). *The reward lies in the family of linear functions $r_\theta(x, y) = \langle \theta, \phi(x, y) \rangle = \theta^\top \phi(x, y)$ for some known and fixed $\phi(x, y) : \mathcal{X} \times \mathcal{Y} \to \mathbb{R}^d$ with $\max_{x,y} \|\phi(x, y)\|_2 \leq 1$. Let $\theta^\star$ be the true parameter for the ground-truth reward function. To ensure the identifiability of $\theta^\star$, we let $\theta^\star \in \Theta_B$, where*

$$\Theta_B = \{\theta \in \mathbb{R}^d \big| \langle 1, \theta \rangle = 0, \|\theta\|_2 \leq B\}. \tag{6}$$

For an LLM, $\phi$ can be considered as the last hidden state of the sequence. In the online RLHF process with several iterations, given a preference dataset $\mathcal{D}_t = \{(x^{(i)}, y_w^{(i)}, y_l^{(i)})\}_{i=1}^n$ at iteration $t$, the reward model is estimated via maximum likelihood estimation, as

$$\hat{\theta}_{\text{MLE}} \in \arg\min_\theta \ell_{\mathcal{D}_t}(\theta), \text{ where } \ell_{\mathcal{D}_t}(\theta) = -\sum_{i=1}^n \log \sigma\left(\langle \theta, \phi(x^{(i)}, y_w^{(i)}) - \phi(x^{(i)}, y_l^{(i)})\rangle\right). \tag{7}$$

When the solution is not unique, we take all of the $\hat{\theta}$ that achieves the minimum. For clarity, we define the expected value function $J_\beta(\pi)$ with MLE estimated reward $\hat{\theta}_{\text{MLE}}$ and $J_\beta^\star(\pi)$ with ground truth reward $\theta^\star$ respectively, as

$$J_\beta(\pi) = \mathbb{E}_{x\sim\rho, y\sim\pi(\cdot|x)}\left[\phi(x,y)^\top\hat{\theta}_{\text{MLE}}\right] - \beta\mathbb{E}_{x\sim\rho}\left[D_{\text{KL}}(\pi(\cdot|x)\|\pi_{\text{ref}}(\cdot|x))\right], \qquad (8)$$

$$J_\beta^\star(\pi) = \mathbb{E}_{x\sim\rho, y\sim\pi(\cdot|x)}\left[\phi(x,y)^\top\theta^\star\right] - \beta\mathbb{E}_{x\sim\rho}\left[D_{\text{KL}}(\pi(\cdot|x)\|\pi_{\text{ref}}(\cdot|x))\right]. \qquad (9)$$

We start with introducing a lemma on bounding the estimation error conditioned on the data $\mathcal{D}_t$.

**Lemma 1.** *(Zhu et al., 2023) For any $\lambda > 0$, letting $\gamma = 1/(2 + e^{-B} + e^B)$ ,with probability at least $1 - \delta$, we have*

$$\|\hat{\theta}_{\text{MLE}} - \theta^\star\|_{\Sigma_{\mathcal{D}_t}+\lambda I} \le C \cdot \sqrt{\frac{d + \log(\frac{1}{\delta})}{\gamma^2 n} + \lambda B^2}, \qquad (10)$$

*where $\Sigma_{\mathcal{D}_t} = \frac{1}{n}\sum_{i=1}^n(\phi(x^{(i)}, y_w^{(i)}) - \phi(x^{(i)}, y_l^{(i)}))(\phi(x^{(i)}, y_w^{(i)}) - \phi(x^{(i)}, y_l^{(i)}))^\top$.*

We refer readers to Zhu et al. (2023) for a detailed proof. Considering a confidence set of parameters

$$\Theta(\hat{\theta}_{\text{MLE}}, \lambda) = \left\{\theta \in \Theta_B \,\big|\, \|\hat{\theta}_{\text{MLE}} - \theta\|_{\Sigma_{\mathcal{D}_t}+\lambda I} \le C \cdot \sqrt{\frac{d + \log(\frac{1}{\delta})}{\gamma^2 n} + \lambda B^2}\right\}, \qquad (11)$$

Lemma 1 shows that with probability at least $1 - \delta$, one has $\theta^\star \in \Theta(\hat{\theta}_{\text{MLE}}, \lambda)$. We thus construct the *optimistic* expected value function $\hat{J}_\beta(\pi)$ which takes the upper confidence bound (UCB) as the reward estimate, as

$$\hat{J}_\beta(\pi; \mathcal{D}_t) = \max_{\theta\in\Theta(\hat{\theta}_{\text{MLE}}, \lambda)} \mathbb{E}_{x\sim\rho, y\sim\pi(\cdot|x)}[\theta^\top(\phi(x,y))] - \beta \cdot \mathbb{E}_{x\sim\rho}[D_{\text{KL}}(\pi(\cdot|x)\|\pi_{\text{ref}}(\cdot|x))]$$

$$= \underbrace{(\mathbb{E}_{x\sim\rho}[\phi(x,\pi(x))])^\top\hat{\theta}_{\text{MLE}} - \beta\mathbb{E}_{x\sim\rho}[D_{\text{KL}}(\pi(\cdot|x)\|\pi_{\text{ref}}(\cdot|x))]}_{\text{original KL-regularized RL tuning objective } J_\beta(\pi)} + \underbrace{\xi\|\mathbb{E}_{x\sim\rho}[\phi(x,\pi(x))]\|_{(\Sigma_{\mathcal{D}_t}+\lambda I)^{-1}}}_{\text{optimistic exploration term (UCB-term)}},$$

$$(12)$$

where $\xi = C\sqrt{\frac{d+\log(1/\delta)}{\gamma^2 n} + \lambda B^2}$. The derivation of Eq. (12) is deferred to Appendix A.1. The first term in Eq. (12) corresponds to the classic two-stage RLHF methods: (i) learning the reward model $\hat{\theta}_{\text{MLE}}$ via MLE under the assumption of BT preference model, and (ii) learning a policy to maximize the estimated reward with KL regularization given $\hat{\theta}_{\text{MLE}}$. The second term, which distinguishes $\hat{J}_\beta(\pi)$ from $J_\beta(\pi)$, is equivalent to a measurement of how well the current dataset covers the distribution of responses generated by the target policy $\pi$. Now we analyze the suboptimality gap of the optimal policy $\hat{\pi}$ derived from optimizing the optimistic expected value function $\hat{J}_\beta(\pi; \mathcal{D}_t)$. For the output policy $\hat{\pi}_t = \arg\max_\pi \hat{J}_\beta(\pi; \mathcal{D}_t)$, we have the following theoretical guarantee.

**Theorem 2.** *For any $\lambda > 0$, $\beta > 0$, with probability at least $1 - \delta$, the optimal policy $\hat{\pi}$ w.r.t the optimistic expected value function $\hat{J}_\beta(\pi; \mathcal{D}_t)$ satisfies*

$$\text{SubOpt}(\hat{\pi}_t) \le 2C \cdot \sqrt{\frac{d + \log(1/\delta)}{\gamma^2 n} + \lambda B^2} \cdot \left\|\mathbb{E}_{x\sim\rho}[\phi(x, \hat{\pi}_t(x))]\right\|_{(\Sigma_{\mathcal{D}_t}+\lambda I)^{-1}}. \qquad (13)$$

The proof is deferred to Appendix A.2. By Theorem 2, we can bound the suboptimality gap of the output policy $\hat{\pi}_t = \arg\max_\pi \hat{J}_\beta(\pi; \mathcal{D}_t)$ for a given iteration $t$. Further, we can analyze how well the policy, resulted from optimizing $\hat{J}_\beta(\pi; \mathcal{D}_t)$ for $T$ iterations in an online RLHF manner, asymptotically converges to the true optimal policy $\pi^\star$. With the total regret after $T$ iterations defined as $\text{Regret}(T) = \sum_{t=1}^T[J_\beta^\star(\pi^\star) - J_\beta^\star(\hat{\pi}_t)]$, we are now ready to state our main theorem which gives a $\tilde{O}(\sqrt{T})$-regret bound in the linear reward setting.

**Theorem 3.** *Assume that for each iteration $1 \le t \le T$, one preference pair sample is collected and added into the dataset in the last iteration $t - 1$. Under Assumption 1, if we set $\lambda = 4$ and denote*

$\iota := \log(1 + (4T)/(d\lambda))$, *with probability at least* $1 - \delta$, *the total regret* $\mathrm{Regret}(T)$ *after* $T$ *iterations satisfies*

$$\mathrm{Regret}(T) \leq \sqrt{T} \cdot C_1 \cdot \sqrt{\frac{d + \log(1/\delta)}{\gamma^2} + \lambda B^2} \cdot \sqrt{d\iota}, \tag{14}$$

*where* $C_1$ *is an absolute constant.*

Theorem 3 shows that adopting an online learning paradigm with the optimistic learning objective $\hat{J}_\beta(\pi; \mathcal{D}_t)$ as the optimization objective achieves at most $\widetilde{O}(\sqrt{T})$ regret, providing a theoretical guarantee for our algorithm implemented based on Eq. (12), where $\tilde{O}$ hides logarithmic dependence on $T$ and $1/\delta$. This analysis can be readily extended to the case where mini-batch samples of size $k$ are collected in every iteration, producing an improved regret bound scaled by $1/\sqrt{k}$. The proof is deferred to the Appendix A.3.

## 3.2 COPO ALGORITHM

In the following, we elaborate on how our COPO actually implements the optimization objective $\hat{J}_\beta(\pi; \mathcal{D}_t)$ in Eq. (12) for LLM alignment. The first term in Eq. (12) involves the same pipeline as the classic RLHF methods: (i) modeling the reward $\hat{\theta}_{\mathrm{MLE}}$ from the preference data $\mathcal{D}_t$, and (ii) fine-tuning the LLM with the estimated reward $\hat{\theta}_{\mathrm{MLE}}$ via RL, which can be integrated into a single direct DPO objective according to Rafailov et al. (2023). Thus, we replace the first term with the objective $\mathcal{L}_{\mathrm{DPO}}(\pi_\varphi; \mathcal{D}_t)$, where the LLM to be optimized is parameterized by $\varphi$. Note that the replacement with DPO objective implies that we implicitly reparameterize the reward function as $r(x, y) = \beta(\log \pi_\varphi(y|x) - \log \pi_{\mathrm{ref}}(y|x))$. It loses the need to design and use the feature mapping $\phi(x, y)$ in the practical implementation while our discussion remains valid in the linear reward setting.

Then, we have the following lemma adapted from Bai et al. (2022a) to build the relationship between the count-based exploration and the second UCB term proportional to $\|\mathbb{E}_{x \sim \rho}[\phi(x, \pi(x))]\|_{(\Sigma_{\mathcal{D}_t} + \lambda I)^{-1}}$.

**Lemma 2.** (Bai et al., 2022a) *In a tabular case where the states and actions are finite, i.e.,* $|\mathcal{X}| < \infty$ *and* $|\mathcal{Y}| < \infty$, *let* $d = |\mathcal{X}||\mathcal{Y}|$ *and* $\phi(x, y) = \mathbf{e}_{(x,y)}$ *be the one-hot canonical basis in* $\mathbb{R}^d$. *The UCB term* $\|\mathbb{E}_{x \sim \rho}[\phi(x, \pi(x))]\|_{(\Sigma_{\mathcal{D}_t} + \lambda I)^{-1}}$ *satisfies*

$$\|\mathbb{E}_{s \sim \rho}[\phi(x, \pi(x))]\|_{(\Sigma_{\mathcal{D}_t} + \lambda I)^{-1}} = \mathbb{E}_{x \sim \rho, y \sim \pi(\cdot|x)}\left[1/\left(\sqrt{N_{\mathcal{D}_t}(x, y) + \lambda}\right)\right], \tag{15}$$

*where* $N_{\mathcal{D}_t}(x, y)$ *is the visit counts of state-action* $(x, y)$ *in the dataset* $\mathcal{D}_t$.

We refer to Bai et al. (2022a; 2024) for a detailed proof. Lemma 2 shows that under the tabular setting, the UCB term takes a simple form as the count-based bonus in the classic RL with exploration (Strehl & Littman, 2008; Bellemare et al., 2016). Combining Lemma 2 and Eq. (12) altogether, we finally derive the optimization objective of COPO, described as

$$\max_{\pi_\varphi} J_{\mathrm{copo}}(\pi_\varphi, \mathcal{D}_t) = -\mathcal{L}_{\mathrm{DPO}}(\pi_\varphi; \mathcal{D}_t, \beta) + \alpha \underbrace{\mathbb{E}_{x \sim \mathcal{D}_t, y \sim \pi_\varphi(y|x)}\left[1/\left(\sqrt{N_{\mathcal{D}_t}(x, y; \vartheta) + \lambda}\right)\right]}_{\text{optimistic term of COPO}}, \tag{16}$$

where $\alpha$ and $\lambda$ are hyperparameters, and $N_{\mathcal{D}_t}(x, y; \vartheta)$ is a counting function with trainable parameter $\vartheta$ discussed in the next section. To demonstrate how our COPO objective implements optimism and elicits active exploration, we analyze its gradient to provide a more intuitive explanation.

**What does the COPO update do?** Denoting the reward function $\hat{r}_\varphi(x, y) = \beta(\log \pi_\varphi(y|x) - \log \pi_{\mathrm{ref}}(y|x))$, the gradient of COPO objective in Eq. (16) with respect to $\varphi$ is derived as follows,

$$\nabla_\varphi J_{\mathrm{copo}}(\pi_\varphi, \mathcal{D}_t) = \beta \mathbb{E}_{(x, y_w, y_l) \sim \mathcal{D}_t}[\sigma(\hat{r}_\varphi(x, y_l) - \hat{r}_\varphi(x, y_w))(\nabla_\varphi \log \pi_\varphi(y_w|x) - \nabla_\varphi \log \pi_\varphi(y_l|x))]$$

$$+ \alpha \cdot \mathbb{E}_{x \sim \mathcal{D}_t, y \sim \pi_{\mathrm{ref}}(y|x)}\left[\frac{\exp(\hat{r}_\varphi(x, y)/\beta)}{\sqrt{N_{\mathcal{D}_t}(x, y; \vartheta) + \lambda}}\nabla_\varphi \log \pi_\varphi(y|x))\right], \tag{17}$$

where $\hat{r}_\varphi(x, y) = \beta \log \pi_\varphi(y|x) - \beta \log \pi_{\mathrm{ref}}(y|x)$ parameterized by DPO reward. We note that the first term remains the same as the original gradient of the DPO loss function, i.e., $-\nabla_\varphi \mathcal{L}_{\mathrm{DPO}}(\pi_\varphi; \mathcal{D})$;

---

**Algorithm 1** Count-based Online Preference Optimization (COPO)

---

**Require:** Reference model $\pi_{\text{ref}}$, preference dataset $\mathcal{D}$, online iterations $T$, optimism coefficient $\alpha$.
1: **for** iteration $t = 1, 2, \ldots, T$ **do**
2:      Set $\tilde{\mathcal{D}}_t$ as the $t$-th portion of $\mathcal{D}$ and generate $y \sim \pi_{\text{ref}}(\cdot \mid x)$ for each prompt $x$ in $\tilde{\mathcal{D}}_t$.
3:      Rank $\{y, y_w, y_l\}$ with score model and obtain $\mathcal{D}_t$ that contains the best and worst responses.
4:      Update the parameter $\vartheta$ via $\min_{f_\vartheta} J_{\text{cfn}}(f_\vartheta; \mathcal{D}_{\text{cfn}})$ for the coin-flipping network via Eq. (18).
5:      Update the LLM policy via $\max_{\pi_\varphi} J_{\text{copo}}(\pi_\varphi; \mathcal{D}_t)$ defined in Eq. (16) and set $\pi_\varphi \to \pi_{\text{ref}}$.
6: **end for**

---

while the second term, corresponding to the gradient of the optimistic term of COPO, controls the optimization direction of LLM according to both the rewards and the visitation counts. Specifically, it tends to increase the log-likelihood of response $y$ generated by $\pi_\varphi$ toward potentially more rewarding areas when its visit counts $N_{\mathcal{D}_t}(x, y)$ in the past is relatively low, rather than those responses with high visit counts. Consequently, the count-based bonus encourages active exploration toward not only high-reward but also more uncertain regions with respect to the regions the LLM has already confirmed, i.e., *optimism in the face of uncertainty* (Agarwal et al., 2019; Lattimore & Szepesvári, 2020; Qiu et al., 2022; Yang et al., 2022). In each update, we maximize the objective in Eq. (16) to make the fine-tuned LLM achieve a trade-off that balances the reward-maximizing and highly uncertain-response pursuing, i.e., the well-known exploration-exploitation trade-off (Mannor & Tsitsiklis, 2004). In each iteration of COPO, the LLM policy will collect novel prompt-response pairs on these uncertain regions, and we use an off-the-shelf reward model to construct $\mathcal{D}_t$. Ideally, with the infinite number of iterations, the datasets $\cup_t \mathcal{D}_t$ collected by the LLM policies can cover the entire prompt-response space with count-based exploration, and the DPO objective aims to find the best policy in such a space with wide data coverage. We give an algorithmic description in Alg. 1.

### 3.3 PSEUDO-COUNT VIA COIN FLIPPING NETWORK

While such a count-based exploration objective in Eq. (16) has theoretical guarantees, it suffers from the issue that visit counts are not directly useful in a large space where same states are rarely visited more than once (Bellemare et al., 2016). There is no doubt that it would be significantly amplified in the prompt-response space of LLMs that is composed of extremely vast discrete token sequences.

Inspired by the count-based exploration in RL (Bellemare et al., 2016; Ostrovski et al., 2017), we can substitute the empirical visit count $N_{\mathcal{D}_t}(x, y)$ with a *pseudo-count* $\hat{N}_{\mathcal{D}_t}(x, y)$ through density models. However, it requires learning-positive properties and powerful neural density models, making them impractical in online RLHF settings. Motivated by the recent progress on estimating pseudo-counts without restrictions on the type of function approximator or the procedure used to train it, we instead simply apply a Coin Flipping Network (CFN) (Lobel et al., 2023) that directly predicts the count-based exploration bonus by solving a simple regression problem.

**The Mechanism underlying CFN.** The key insight in the CFN is that a state's visitation count can be derived from the sampling distribution of Rademacher trials (or *coin flips*) made every time a state is encountered. The CFN $f_\vartheta$ parameterized by $\vartheta$ is learned by solving $\arg\min_\vartheta \mathbb{E}_{(s_i, s_i^{\text{label}}) \sim \mathcal{D}_{\text{cfn}}}[\mathcal{L}(s_i, s_i^{\text{label}})]$ where $\mathcal{L}$ is the mean-square error loss function and $\mathcal{D}_{\text{cfn}}$ is a dataset of state-label pairs for learning the CFN. In our case, the state $s$ is the feature vector of prompt-response pair $(x, y)$. Considering the fair coin-flip distribution $\mathcal{C}$ over outcomes $\{-1, 1\}$, if we flip the coin $m$ times and average the results into $z_m$, the second moment of $z_m$ is related to the inverse count: $\mathcal{M}_2(z_m) = \mathbb{E}[z_m^2] = \sum_i \Pr(z_m = i) * i^2 = 1/m$. Furthermore, by flipping $d$ coins each time, the variance of $z_m^2$ can be reduced by a factor of $\frac{1}{d}$, which implies a reliable way for estimating the inverse count (Lobel et al., 2023). To this end, we generate a $d$-dimensional random vector $\mathbf{c_i} \sim \{-1, 1\}^d$ as a label $s_i^{\text{label}}$ for state $s_i$. The learning objective is described as

$$\min_{f_\vartheta} J_{\text{cfn}}(f_\vartheta; \mathcal{D}_{\text{cfn}}) = \mathbb{E}_{(s_i, s_i^{\text{label}}) \sim \mathcal{D}_{\text{cfn}}}[\mathcal{L}(s_i, s_i^{\text{label}})] = \arg\min_\vartheta \sum_{i=1}^{|\mathcal{D}_{\text{cfn}}|} \|\mathbf{c_i} - f_\vartheta(s_i)\|^2, \quad (18)$$

where $f_\vartheta(s_i)$ is a neural network that extracts the feature vectors of prompt-response pairs. In practice, we adopt $s_i$ as the last hidden state of the fixed LLM with the prompt-response pair as input, and $f_\vartheta(\cdot)$ is set to a lightweight network with several fully-connected layers.

The dataset $\mathcal{D}_{\mathrm{cfn}}$ is constructed by using prompt from $\mathcal{D}_t$ and responses generated by the LLM policy in the previous iteration, where each occurrence of a state is paired with a different random vector. In a case where there are $m$ instances of the same state $s_i$ in $\mathcal{D}_{\mathrm{cfn}}$, the optimal solution $f_\vartheta^*$ satisfies $f_\vartheta^*(s_i) = \frac{1}{m} \sum_{i=1}^m \mathbf{c_i}$ according to Lobel et al. (2023), then the reciprocal pseudo-count can be estimated by

$$\frac{1}{d}\|f_\vartheta(s)\|^2 = \frac{1}{d}\sum_{j=1}^d \mathbb{E}\Big[\Big(\sum_{i=1}^m \frac{c_{ij}}{m}\Big)\Big] = \frac{1}{d}\sum_{j=1}^d \mathbb{E}\Big[z_m^2\Big] = \frac{1}{m}. \tag{19}$$

Thus, by training $f_\vartheta$ on the objective shown in Eq. (18) we can simply approximate the count-based bonus given by $\sqrt{\|f_\vartheta(s)\|^2/d} \approx \sqrt{1/N(s)}$.

## 4  RELATED WORKS

**RLHF and iterative online RLHF.**  The RLHF framework used for aligning LLMs was first introduced in Christiano et al. (2017); Ziegler et al. (2019) and further developed in Instruct-GPT (Ouyang et al., 2022), LLaMA-2 (Touvron et al., 2023) and etc. These works share a similar pipeline that is typically made up of two separate stages: estimating a reward model based on the BT model (Bradley & Terry, 1952) and using PPO (Schulman et al., 2017) to optimize the reward signals together with a KL regularization. Several efforts have been made to simplify the preference alignment procedure and improve the performance of RLHF (Zhao et al., 2023; Rafailov et al., 2023; Munos et al., 2023; Azar et al., 2024; Guo et al., 2024; Swamy et al., 2024; Tang et al., 2024; Ethayarajh et al., 2024b). According to whether preference data is collected before training or by using the current policy during training, we can roughly divide these methods into two categories: offline RLHF and (iterative) online RLHF. In offline RLHF, a line of work studies direct preference learning, including DPO (Rafailov et al., 2023) and its variants (Xu et al., 2024; Lee et al., 2024a; Rafailov et al., 2024b). These algorithms integrate reward modeling and RL-tuning into a single policy objective and optimize it directly on the offline preference dataset. It is observed that DPO-based algorithms are more stable than PPO (Tunstall et al., 2023; Dubois et al., 2024) and have also been adopted in preference learning for other RL problems (Yuan et al., 2024b; Yu et al., 2024).

On the other hand, (iterative) online RLHF means that we can collect extra responses by sampling responses from the LLM itself and querying preference feedback from humans or AI. This strategy can help mitigate the OOD issue of the learned reward model (Gao et al., 2023) and gradually push beyond the boundary of human capabilities. In online RLHF, online exploration is crucial to increase the coverage of preference data that determines policy improvement. There are several works proposing various techniques to encourage exploration for online RLHF. Dwaracherla et al. (2024) proposed using the posterior of reward models to approximately measure the uncertainty for active exploration. XPO (Xie et al., 2024) leveraged the property of the approximation of the regularized value function under the token-level MDP formulation with general function approximation. Similar to ours, SELM (Zhang et al., 2024) and VPO (Cen et al., 2024) considered the optimism from perspective of learning reward model, but achieved it by incorporating the maximum of the KL-regularized value function over the target LLM $\max_\pi J_\beta(\pi)$ into the reward modeling. Instead, our COPO implements optimism based on the confidence set of the reward MLE, which is provably efficient and equivalent to count-based exploration in special cases. Other online RLHF works (Yuan et al., 2024a; Lee et al., 2024b; Singhal et al., 2024) study how to automatically annotate preference labels for generated response pairs, while we adopt an off-the-shelf reward model to rank responses.

**Count-based exploration.**  In both bandit and online RL, a promising strategy for exploration is to incorporate a bonus to encourage the agent to gather informative data (Hao et al., 2023), which can be calculated based on count (Strehl & Littman, 2008; Bellemare et al., 2016), prediction error (Pathak et al., 2017), or random network distillation (RND, Burda et al. (2018)). In theoretical RL, count-based exploration is provably efficient in tabular and linear MDPs (Strehl & Littman, 2008; Jin et al., 2020), which motivates us to focus on count-based exploration in online RLHF. In deep RL with large state space, count-based exploration can be extended to function approximation by using density models to calculate pseudo-counts (Bellemare et al., 2016; Bai et al., 2021b;a). However, with these density-based pseudo-counts come many restrictions that are challenging to fulfill. Other methods (Tang et al., 2017; Rashid et al., 2020) instead heavily incorporated domain knowledge to eliminate the usage of density models. In contrast, CFN (Lobel et al., 2023) takes raw states as input and yields a visitation count when optimized for a supervised learning objective.

## 5 EXPERIMENTS

### 5.1 EXPERIMENT SETUP

**Dataset and Ranker** For preference alignment of LLMs, we select UltraFeedback 60K (Cui et al., 2023) that contains preference pairs of single-turn conversation as the preference dataset $\mathcal{D} = \{x, y_w, y_l\}$. For $t$-iteration of online preference alignment, we generate response $y$ for each prompt $x$ in $\tilde{\mathcal{D}}_t$ with the updated LLM, where $\tilde{\mathcal{D}}_t$ is the $t$-th portion of the whole dataset $\mathcal{D}$. Then we adopt a small-sized PairRM (0.4B) model (Jiang et al., 2023) to rank $(y, y_w, y_l)$ and update $\tilde{\mathcal{D}}_t$ to contain the best (chosen) and worst (rejected) responses according to the reward model, denoted as $\mathcal{D}_t$. Finally, we use $\mathcal{D}_t$ for preference alignment with DPO and count-based exploration. In the next iteration, we use the updated LLM to generate a response and construct $\mathcal{D}_{t+1}$ accordingly. We note that the performance of our method can be further improved by using the state-of-the-art reward models in RewardBench (Lambert et al., 2024), while we adopt a small-scale PairRM model for proof-of-concept verification and a fair comparison with baselines.

**Implementation of CFN** We calculate the pseudo-count of the prompt-response pair via CFN. We implement CFN as a small fully-connected network that contains two hidden layers with 32 and 20 units, respectively. CFN takes the last hidden state of the prompt-response pair extracted by a backbone LLM as the state, representing $\phi(x, y)$ in our theoretical analysis. Then FCN uses the state vector to calculate the pseudo-counts via $f_\vartheta(\phi(x, y))$. In the training of CFN, the parameters of backbone LLM are kept fixed, thus we only require a small amount of computation to update the parameter of CFN, which counts the prompt-response pairs. The CFN network is trained with $\mathcal{D}_{\mathrm{cfn}}$ in each iteration and is used to encourage exploration in LLM update with DPO objectives.

**Baselines.** We adopt an online version of DPO (Rafailov et al., 2023) as a baseline, where DPO is trained on $\mathcal{D}_t$ that contains responses of the updated LLM policy. We also adopt SELM (Zhang et al., 2024) as the state-of-the-art online RLHF algorithm, which performs exploration towards potentially high-reward responses without considering the confidence of LLM in these responses. We adopt the same hyper-parameter settings of online DPO and SELM as in Zhang et al. (2024), where the algorithms are trained under the best hyper-parameter setting via a grid search. Both SELM and online DPO are finetuned based on the SFT model. For a comprehensive evaluation, we adopt Zephyr (Tunstall et al., 2023) and Llama-3 (Meta, 2024) for RLHF alignment. Specifically, we choose Zephyr-7B-SFT with a single iteration of standard DPO training on the first portion of the training data, which is the same as SELM. And we directly perform preference alignment for Llama-3-8B-Instruct that has been tuned through SFT. For both Zephyr-7B-SFT and Llama-3-8B-Instruct, we conduct 3 iterations of DPO/SELM/COPO alignment of training for comparison.

### 5.2 EXPERIMENT RESULTS

We evaluate our method on instruction-following benchmark AlpacaEval 2.0 (Dubois et al., 2024) and MT-Bench (Zheng et al., 2023). AlpacaEval addresses the consistency of results by using a standardized process for comparing model outputs to reference responses. The evaluation set, AlpacaFarm, while diverse, is designed to test models across a broad range of simple instructions, ensuring a consistent benchmark for model performance. According to the results of AlpacaEval 2.0 in Table 1, we find COPO increases the LC win rate of AlpacaEval 2.0 from 22.19 to 27.21 for Zephyr-7B, and increases the LC win rate from 33.17 to 35.54 for Llama3-8B-It, which is a significant improvement in instruction-following tasks. The result signifies that the count-based objective enhances the exploration ability of the LLM, which results in better coverage of the underlying prompt-response space. Thus, the LLM policy obtains datasets with better coverage on the optimal prompt-response pairs and benefits the policy optimization of LLMs. As we discussed in the theoretical part, the UCB-based exploration reduces the suboptimality gap in preference optimization, and the empirical result verifies the theoretical results. Regarding COPO results with Llama-3-8B-Instruct, we find that the proposed iterative algorithm armed with a count-based exploration term can even outperform much larger LLMs, such as Yi-34B-Chat (Young et al., 2024) and Llama-3-70B-Instruct (Dubey et al., 2024) in LC win rate. The evaluation results in MT-Bench show similar performance improvement compared to online DPO and outperform SELM, where COPO adopts pseudo-count for weighting in policy update compared to SELM. The results in MT-Bench outperform Yi-34B-Chat while still inferior to Llama-3-70B-Instruct.

| Model | AlpacaEval 2.0 | | | MT-Bench | | |
|---|---|---|---|---|---|---|
| | LC Win Rate | Win Rate | Avg. len | Avgerage | 1st Turn | 2nd Turn |
| Zephyr-7B-SFT | 8.01 | 4.63 | 916 | 5.30 | 5.63 | 4.97 |
| Zephyr-7B-DPO | 15.41 | 14.44 | 1752 | 7.31 | 7.55 | 7.07 |
| DPO Iter 1 (Zephyr) | 20.53 | 16.69 | 1598 | 7.53 | 7.81 | 7.25 |
| DPO Iter 2 (Zephyr) | 22.12 | 19.82 | 1717 | 7.55 | 7.85 | 7.24 |
| DPO Iter 3 (Zephyr) | 22.19 (↑14.18) | 19.88 (↑15.25) | 1717 | 7.46 (↑2.16) | 7.85 | 7.06 |
| SELM Iter 1 (Zephyr) | 20.52 | 17.23 | 1624 | 7.53 | 7.74 | 7.31 |
| SELM Iter 2 (Zephyr) | 21.84 | 18.78 | 1665 | 7.61 | 7.85 | 7.38 |
| SELM Iter 3 (Zephyr) | 24.25 (↑16.24) | 21.05 (↑16.42) | 1694 | 7.61 (↑2.31) | 7.74 | 7.49 |
| COPO Iter 1 (Zephyr) | 26.43 | 21.61 | 1633 | 7.68 | 7.72 | 7.64 |
| COPO Iter 2 (Zephyr) | **27.21** (↑19.20) | 22.61 | 1655 | 7.78 | 7.85 | 7.71 |
| COPO Iter 3 (Zephyr) | 26.91 | **23.60** (↑18.97) | 1739 | **7.79** (↑2.49) | 7.89 | 7.69 |
| Llama-3-8B-Instruct | 22.92 | 22.57 | 1899 | 7.93 | 8.47 | 7.38 |
| DPO Iter 1 (Llama3-It) | 30.89 | 31.60 | 1979 | 8.07 | 8.44 | 7.70 |
| DPO Iter 2 (Llama3-It) | 33.91 | 32.95 | 1939 | 7.99 | 8.39 | 7.60 |
| DPO Iter 3 (Llama3-It) | 33.17 (↑10.25) | 32.18 (↑9.61) | 1930 | 8.18 (↑0.25) | 8.60 | 7.77 |
| SELM Iter 1 (Llama3-It) | 31.09 | 30.90 | 1956 | 8.09 | 8.57 | 7.61 |
| SELM Iter 2 (Llama3-It) | 33.53 | 32.61 | 1919 | 8.18 | 8.69 | 7.66 |
| SELM Iter 3 (Llama3-It) | 34.67 (↑11.75) | **34.78** (↑12.21) | 1948 | 8.25 (↑0.32) | 8.53 | 7.98 |
| COPO Iter 1 (Llama3-It) | 33.68 | 33.15 | 1959 | 8.12 | 8.38 | 7.86 |
| COPO Iter 2 (Llama3-It) | 34.30 | 33.31 | 1939 | 8.25 | 8.49 | 8.01 |
| COPO Iter 3 (Llama3-It) | **35.54** (↑12.62) | 32.94 (↑10.37) | 1930 | **8.32** (↑0.39) | 8.53 | 8.11 |
| SPIN | 7.23 | 6.54 | 1426 | 6.54 | 6.94 | 6.14 |
| Orca-2.5-SFT | 10.76 | 6.99 | 1174 | 6.88 | 7.72 | 6.02 |
| DNO (Orca-2.5-SFT) | 22.59 | 24.97 | 2228 | 7.48 | 7.62 | 7.35 |
| Mistral-7B-Instruct-v0.2 | 19.39 | 15.75 | 1565 | 7.51 | 7.78 | 7.25 |
| SPPO (Mistral-it) | 28.53 | 31.02 | 2163 | 7.59 | 7.84 | 7.34 |
| Yi-34B-Chat | 27.19 | 21.23 | 2123 | 7.90 | - | - |
| Llama-3-70B-Instruct | 33.17 | 33.18 | 1919 | 9.01 | 9.21 | 8.80 |
| GPT-4 Turbo (04/09) | 55.02 | 46.12 | 1802 | 9.19 | 9.38 | 9.00 |

Table 1: Results on AlpacaEval 2.0 and MT-Bench. The red arrows indicate the increment from the SFT model (i.e., Zephyr-7B-DPO and Llama-3-8B-Instruct). Compared to online DPO and online SELM baselines, our method achieves superior performance and is competitive with larger models.

According to the result in AlpacaEval, our method generally increases performance after each iteration in the online RLHF process. However, in Zephyr experiments, we find the average length of response increases significantly in the last iterations, resulting in a decreased LC win rate compared to previous iterations. This problem can be caused by the proposed exploration term, which can encourage the LLM policy to generate longer sentences that can be novel compared to previous responses. Therefore, a future direction is to combine our method with existing length control methods to reduce such a bias (Meng et al., 2024; Singhal et al., 2023).

We evaluate our approach and the baseline models using established academic benchmarks from the LLM leaderboard (Gao et al., 2024), such as GSM8K (Cobbe et al., 2021), HellaSwag (Zellers et al., 2019), ARC challenge (Clark et al., 2018), TruthfulQA (Lin et al., 2021), EQ-Bench (Paech, 2023), and OpenBookQA (OBQA) (Mihaylov et al., 2018). Following the settings in SELM (Zhang et al., 2024), we employ various Chain of Thought (CoT) configurations, including zero-shot and few-shot scenarios. Table 2 presents the results for these benchmarks. Furthermore, our method exhibits improved performance on these academic benchmarks after incorporating preference alignment on language tasks. Our technique achieves a suitable balance between maximizing rewards and exploring the response space without compromising the reasoning accuracy in academic tasks. However, since the preference dataset focuses on how well the model follows instructions or completes tasks as intended by humans, such an alignment process might not necessarily align well with the requirements of some academic benchmarks (e.g., ARC), which often require abstract reasoning, complex inference, or extensive factual knowledge that may not be enhanced by instruction following.

## 5.3 ABLATION STUDY

We conduct an ablation study on Llama model to show the effectiveness of the proposed exploration term in COPO. Table 3 shows the results of the evaluation on AlpacaEval with different factors $\alpha$ of the exploration terms. According to the results, choosing a suitable $\alpha$ is important to balance exploration and preference alignment. We visualize the count-based term (i.e., $\mathbb{E}_{x,y \in \mathcal{D}_t} 1/\sqrt{N(x,y) + \lambda}$)

| Models | GSM8K (8-s CoT) | HellaSwag (10-s) | ARC (25-s) | TruthfulQA (0-s) | EQ (0-s) | OBQA (10-s) | Average |
|---|---|---|---|---|---|---|---|
| Zephyr-7B-SFT | 43.8 | 82.2 | 57.4 | 43.6 | 39.1 | 35.4 | 50.3 |
| Zephyr-7B-DPO | 47.2 | 84.5 | 61.9 | 45.5 | 65.2 | 38.0 | 57.0 |
| DPO Iter 1 (Zephyr) | 45.5 | 85.2 | 62.1 | 52.4 | 68.4 | 39.0 | 58.8 |
| DPO Iter 2 (Zephyr) | 44.9 | 85.4 | 62.0 | 53.1 | 69.3 | 39.4 | 59.0 |
| DPO Iter 3 (Zephyr) | 43.2 | 85.2 | 60.8 | 52.5 | 69.1 | 39.6 | 58.4 |
| SELM Iter 1 (Zephyr) | 46.3 | 84.8 | 62.9 | 52.9 | 68.8 | 39.6 | 59.2 |
| SELM Iter 2 (Zephyr) | 46.2 | 85.4 | 62.1 | 53.1 | 69.3 | 39.6 | 59.3 |
| SELM Iter 3 (Zephyr) | 43.8 | 85.4 | 61.9 | 52.4 | 69.9 | 39.8 | 58.9 |
| COPO Iter 1 (Zephyr) | 46.8 | 85.0 | 62.4 | 53.0 | 68.7 | 39.3 | 59.2 |
| COPO Iter 2 (Zephyr) | 46.7 | 85.3 | 62.5 | 53.3 | 69.1 | 39.8 | 59.5 |
| COPO Iter 3 (Zephyr) | 47.0 | 85.4 | 62.9 | 53.4 | 69.9 | 40.3 | 59.9 |
| Llama-3-8B-Instruct | 76.7 | 78.6 | 60.8 | 51.7 | 61.8 | 38.0 | 61.3 |
| DPO Iter 1 (Llama3-It) | 78.5 | 81.7 | 63.9 | 55.5 | 64.1 | 42.6 | 64.4 |
| DPO Iter 2 (Llama3-It) | 79.4 | 81.7 | 64.4 | 56.4 | 64.3 | 42.6 | 64.8 |
| DPO Iter 3 (Llama3-It) | 80.1 | 81.7 | 64.1 | 56.5 | 64.1 | 42.6 | 64.8 |
| SELM Iter 1 (Llama3-It) | 78.7 | 81.7 | 64.5 | 55.4 | 64.1 | 42.4 | 64.5 |
| SELM Iter 2 (Llama3-It) | 79.3 | 81.8 | 64.7 | 56.5 | 64.2 | 42.6 | 64.9 |
| SELM Iter 3 (Llama3-It) | 80.1 | 81.8 | 64.3 | 56.5 | 64.2 | 42.8 | 65.0 |
| COPO Iter 1 (Llama3-It) | 79.1 | 81.7 | 64.3 | 56.4 | 64.3 | 43.0 | 64.8 |
| COPO Iter 2 (Llama3-It) | 79.3 | 81.8 | 64.6 | 56.4 | 64.4 | 43.2 | 65.0 |
| COPO Iter 3 (Llama3-It) | 80.2 | 81.8 | 64.7 | 56.5 | 64.4 | 43.6 | 65.2 |
| SPIN | 44.7 | 85.9 | 65.9 | 55.6 | 54.4 | 39.6 | 57.7 |
| Mistral-7B-Instruct-v0.2 | 43.4 | 85.3 | 63.4 | 67.5 | 65.9 | 41.2 | 61.1 |
| SPPO (Mistral-it) | 42.4 | 85.6 | 65.4 | 70.7 | 56.5 | 40.0 | 60.1 |

Table 2: Performance comparison between COPO and the baselines on academic multi-choice QA benchmarks in standard zero-shot, few-shot, and CoT settings. Here, n-s refers to n-shot. The red and blue texts represent the best and the second-best results.

in Fig. 1, where the steps contain 3 iterations split by a dashed line. After each iteration, we collect new responses on the prompt set and reupdate the CFN network. The result shows that a large $\alpha$ encourages the LLM to collect novel responses in the first two iterations. After preference alignment, the optimism term in the last iteration decreases as the policy has converged to a local-optimal solution and the performance will not increase in the last iteration. For a small $\alpha$, the LLM policy gradually collects novel responses in each iteration, while the final performance is limited by its exploration ability. A suitable $\alpha$ enables the policy to focus on exploration in early iterations and preference alignment in later iterations, resulting in better final performance.

| Factor | Iter1 | Iter2 | Iter3 |
|---|---|---|---|
| 0.01 | 32.89 | 33.18 | 33.76 |
| 0.10 | 33.68 | 34.30 | 35.54 |
| 0.50 | 33.70 | 34.61 | 34.82 |

Table 3: COPO results on AlpacaEval 2.0 with different exploration factor $\alpha$ in 3 iterations.

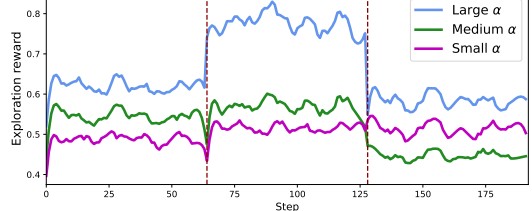

Figure 1: Exploration rewards in 3 iterations with different $\alpha$.

## 6 CONCLUSION

This paper presents COPO, a novel algorithm for online RLHF of LLMs. COPO integrates count-based exploration with the RLHF framework, achieving a tight regret bound policy through the use of an UCB bonus. COPO can balance exploration and preference optimization via a lightweight pseudo-counting module, and obtains superior performance in AlphaEval 2.0, MT-Bench, and LLM leaderboard evaluations compared to other leading online RLHF algorithms. A future direction is to further enhance the exploration ability of our method by using a changing prompt set, avoiding the restrictions on the initial dataset and resulting in better coverage on the prompt-response space. Meanwhile, an automatic tuned exploration factor according to the status of the LLM policy and the collected data would be better to balance exploration and alignment in different iterations.

## REPRODUCIBILITY STATEMENT

For the theoretical part, we provide the detailed theoretical proof in Appendix A. For the practical part, we give experiment setup in Section 5. The hyper-parameters and implementation details are given in Appendix B. The code is released at `https://github.com/Baichenjia/COPO`.

## ACKNOWLEDGMENTS

This work is supported by National Key Research and Development Program of China (Grant No.2024YFE0210900) and National Natural Science Foundation of China (Grant No.62306242).

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

# A  THEORETICAL PROOF

## A.1  DERIVATION OF EQUATION (12)

*Proof.* For any $\theta \in \Theta(\hat{\theta}_{\mathrm{MLE}}, \lambda)$, by Cauchy-Schwartz inequality, the following holds,

$$|(\mathbb{E}_{x \sim \rho}[\phi(x, \pi(x))])^{\top}(\theta - \hat{\theta}_{\mathrm{MLE}})| \le \|\theta - \hat{\theta}_{\mathrm{MLE}}\|_{\Sigma_{\mathcal{D}_t} + \lambda I} \cdot \|\mathbb{E}_{x \sim \rho}[\phi(x, \pi(x))]\|_{(\Sigma_{\mathcal{D}_t} + \lambda I)^{-1}}$$

$$\le C \cdot \sqrt{\frac{d + \log(\frac{1}{\delta})}{\gamma^2 n} + \lambda B^2} \cdot \|\mathbb{E}_{x \sim \rho}[\phi(x, \pi(x))]\|_{(\Sigma_{\mathcal{D}_t} + \lambda I)^{-1}}$$

Thus, we have

$$(\mathbb{E}_{x \sim \rho}[\phi(x, \pi(x))])^{\top}\theta \le (\mathbb{E}_{x \sim \rho}[\phi(x, \pi(x))])^{\top}\hat{\theta}_{\mathrm{MLE}} + \xi \cdot \|\mathbb{E}_{x \sim \rho}[\phi(x, \pi(x))]\|_{(\Sigma_{\mathcal{D}_t} + \lambda I)^{-1}} \quad (20)$$

A feasible way to maximize $\mathbb{E}_{x \sim \rho}[\theta^{\top}\phi(x, \pi(x))]$ over the confidence set $\Theta(\hat{\theta}_{\mathrm{MLE}}, \lambda)$ is to choose the RHS of Eq. (20) as the maximum. Q.E.D.  $\square$

## A.2  PROOF OF THEOREM 2

*Proof.* Note that the optimistic expected value function at iteration $t$ is

$$\hat{J}_{\beta}(\pi; \mathcal{D}_t) = \max_{\theta \in \Theta(\hat{\theta}_{\mathrm{MLE}}, \lambda)} \mathbb{E}_{x \sim \rho, y \sim \pi(\cdot|x)}[\theta^{\top}(\phi(x, y))] - \beta \cdot \mathbb{E}_{x \sim \rho}[D_{\mathrm{KL}}(\pi(\cdot|x)\|\pi_{\mathrm{ref}}(\cdot|x))].$$

For ease of presentation, we define

$$J_{\beta}^{\star}(\pi) = \mathbb{E}_{x \sim \rho, y \sim \pi(\cdot|x)}[r_{\theta^{\star}}(x, y)] - \beta \cdot \mathbb{E}_{x \sim \rho}[D_{\mathrm{KL}}(\pi(\cdot|x)\|\pi_{\mathrm{ref}}(\cdot|x))]$$

$$= \mathbb{E}_{x \sim \rho}[\theta^{\star\top}\phi(x, \pi(x))] - \beta \cdot \mathbb{E}_{x \sim \rho}[D_{\mathrm{KL}}(\pi(\cdot|x)\|\pi_{\mathrm{ref}}(\cdot|x))],$$

and omit the $\mathcal{D}_t$ in $\hat{J}_{\beta}(\pi; \mathcal{D}_t)$, i.e., setting $\hat{J}_{\beta}(\pi) := \hat{J}_{\beta}(\pi; \mathcal{D}_t)$.

According to Lemma 1, we know that $\theta^{\star} \in \Theta(\hat{\theta}_{\mathrm{MLE}}, \lambda)$ with probability at least $1 - \delta$. Consequently, we have

$$\hat{J}_{\beta}(\pi) \ge J_{\beta}^{\star}(\pi), \text{ for any } \pi \quad (21)$$

Meanwhile, by $\hat{\pi}_t = \arg\max_{\pi} \hat{J}_{\beta}(\pi)$ we also have

$$\hat{J}_{\beta}(\hat{\pi}_t) \ge \hat{J}_{\beta}(\pi^{\star}) \quad (22)$$

Combining Eq. (21) and Eq. (22), we have

$$\hat{J}_{\beta}(\hat{\pi}_t) \ge \hat{J}_{\beta}(\pi^{\star}) \ge J_{\beta}^{\star}(\pi^{\star})$$

Substituting this into the definition of the suboptimality gap, we achieve

$$\mathbf{SubOpt}(\hat{\pi}_t) = J_{\beta}^{\star}(\pi^{\star}) - J_{\beta}^{\star}(\hat{\pi}_t) \le \hat{J}_{\beta}(\hat{\pi}_t) - J_{\beta}^{\star}(\hat{\pi}_t)$$

Here we need to introduce a necessary notation of $\hat{\theta}_t$:

$$\hat{\theta}_t = \arg\max_{\theta \in \Theta(\hat{\theta}_{\mathrm{MLE}}, \lambda)} \{(\mathbb{E}_{x \sim \rho}[\phi(x, \hat{\pi}_t(x))])^{\top}\theta - \beta \cdot \mathbb{E}_{x \sim \rho}[D_{\mathrm{KL}}(\hat{\pi}_t(\cdot|x)\|\pi_{\mathrm{ref}}(\cdot|x))]\}$$

With the extra notation, we can further have

$$\mathbf{SubOpt}(\hat{\pi}_t) \le \hat{J}_{\beta}(\hat{\pi}_t) - J_{\beta}^{\star}(\hat{\pi}_t)$$

$$= \mathbb{E}_{x \sim \rho}[(\hat{\theta}_t - \theta^{\star})^{\top}\phi(x, \hat{\pi}_t(x))]$$

$$= \mathbb{E}_{x \sim \rho}[(\hat{\theta}_t - \hat{\theta}_{\mathrm{MLE}})^{\top}\phi(x, \hat{\pi}_t(x))] + \mathbb{E}_{x \sim \rho}[(\hat{\theta}_{\mathrm{MLE}} - \theta^{\star})^{\top}\phi(x, \hat{\pi}_t(x))]$$

$$\le (\|\hat{\theta}_t - \hat{\theta}_{\mathrm{MLE}}\|_{\Sigma_{\mathcal{D}_t} + \lambda I} + \|\hat{\theta}_{\mathrm{MLE}} - \theta^{\star}\|_{\Sigma_{\mathcal{D}_t} + \lambda I}) \cdot \|\mathbb{E}_{x \sim \rho}[\phi(x, \hat{\pi}_t(x))]\|_{(\Sigma_{\mathcal{D}_t} + \lambda I)^{-1}}$$

$$\le 2C \cdot \sqrt{\frac{d + \log(1/\delta)}{\gamma^2 n} + \lambda B^2} \cdot \|\mathbb{E}_{x \sim \rho}[\phi(x, \hat{\pi}_t(x))]\|_{(\Sigma_{\mathcal{D}_t} + \lambda I)^{-1}},$$

where the second inequality uses Cauchy-Schwarz inequality, and the last inequality is obtained by Lemma 1.  $\square$

### A.3    PROOF OF THEOREM 3

We first present an important lemma and a corollary for a special case. Then, we combine the lemma and corollary to prove Theorem 3.

**Lemma 3.** (Abbasi-Yadkori et al., 2011). *Let $\{\phi_t\}_{t \geq 0}$ be a bounded sequence in $\mathbb{R}^d$ satisfying $\sup_{t \geq 0} \|\phi_t\| \leq 1$. Let $\Lambda_0 \in \mathbb{R}^{d \times d}$ be a positive definite matrix. For any $t \geq 0$, we define $\Lambda_t = \Lambda_0 + \sum_{j=1}^t \phi_j \phi_j^\top$. Then, if the smallest eigenvalue of $\Lambda_0$ satisfies $\lambda_{min}(\lambda_0) \geq 1$, we have*

$$\log \left[ \frac{\det(\Lambda_t)}{\det(\Lambda_0)} \right] \leq \sum_{j=1}^t \phi_j^\top \Lambda_{j-1}^{-1} \phi_j \leq 2 \log \left[ \frac{\det(\Lambda_t)}{\det(\Lambda_0)} \right]. \tag{23}$$

*Proof.* The following proof refers mostly to Jin et al. (2020). Since $\lambda_{\min}(\Lambda_0) \geq 1$ and $\|\phi_t\| \leq 1$ for all $j \geq 0$, we have

$$\phi_j^\top \Lambda_{j-1}^{-1} \phi_j \leq [\lambda_{\min}(\Lambda_0)]^{-1} \cdot \|\phi_j\|^2 \leq 1, \quad \forall j \geq 0.$$

For any $x \in [0,1]$, it holds that $\log(1+x) \leq x \leq 2\log(1+x)$. Therefore, we have

$$\sum_{j=1}^t \log(1 + \phi_j^\top \Lambda_{j-1}^{-1} \phi_j) \leq \sum_{j=1}^t \phi_j^\top \Lambda_{j-1}^{-1} \phi_j \leq 2 \sum_{j=1}^t \log(1 + \phi_j^\top \Lambda_{j-1}^{-1} \phi_j) \tag{24}$$

Elementary algebra gives

$$\det(\Lambda_t) = \det(\Lambda_{t-1} + \phi_t \phi_t^\top) = \det(\Lambda_{t-1}) \det(I + \Lambda_{t-1}^{-\frac{1}{2}} \phi_t \phi_t^\top \Lambda_{t-1}^{-\frac{1}{2}})$$

$$= \det(\Lambda_{t-1})(1 + \phi_t^\top \Lambda_{t-1}^{-1} \phi_t) = \det(\Lambda_0) \prod_{i=1}^t (1 + \phi_i^\top \Lambda_{i-1}^{-1} \phi_i).$$

Hence, we have

$$\sum_{j=1}^t \log(1 + \phi_j^\top \Lambda_{j-1}^{-1} \phi_j) = \log \det(\Lambda_t) - \log \det(\Lambda_0) \tag{25}$$

Combining Eq. (24) and Eq. (25), we conclude the proof of Eq. (23). □

**Corollary 1.** *Under the same setting of Lemma 3, if the smallest eigenvalue of $\Lambda_0$ satisfies $\lambda_{min}(\lambda_0) \geq 4$, for reference vectors $\{\nu_j\}_{j=1}^t \in \mathbb{R}^d$ which are also bounded: $\|\nu_j\| \leq 1$ for any $j$, we have*

$$\log \left[ \frac{\det(\Lambda_t)}{\det(\Lambda_0)} \right] \leq \sum_{j=1}^t (\phi_j + \nu_j)^\top \Lambda_{j-1}^{-1} (\phi_j + \nu_j) \leq 2 \log \left[ \frac{\det(\Lambda_t)}{\det(\Lambda_0)} \right]. \tag{26}$$

*Proof.* By triangle inequality, we have

$$(\phi_j + \nu_j)^\top \Lambda_{j-1}^{-1} (\phi_j + \nu_j) \leq [\lambda_{\min}(\Lambda_0)]^{-1} \cdot \|\phi_j + \nu_j\|^2 \leq [\lambda_{\min}(\Lambda_0)]^{-1} \cdot 2(\|\phi_j\|^2 + \|\nu_j\|^2) \leq 1$$

when $\lambda_{\min}(\Lambda_0) \geq 4$ and $\|\nu\| \leq 1$. Then we consider the determinant of the updated matrix $\Lambda_t$:

$$\det(\Lambda_t) = \det(\Lambda_{t-1} + (\phi_t + \nu_t)(\phi_t + \nu_t)^\top)$$
$$= \det(\Lambda_{t-1}) \det(I + \Lambda_{t-1}^{-\frac{1}{2}} (\phi_t + \nu_t)(\phi_t + \nu_t)^\top \Lambda_{t-1}^{-\frac{1}{2}}). \tag{27}$$

Applying the matrix determinant lemma, which states that for any invertible matrix $A$ and vectors $u$ and $v$, we have $\det(A + uv^\top) = \det(A)(1 + v^\top A^{-1} u)$. Then we can simplify the expression as follows:

$$\det(\Lambda_t) = \det(\Lambda_{t-1})(1 + (\phi_t + \nu_t)^\top \Lambda_{t-1}^{-1} (\phi_t + \nu_t)) \tag{28}$$

By recursively applying this step, we obtain:

$$\det(\Lambda_t) = \det(\Lambda_0) \prod_{j=1}^{t} (1 + (\phi_j + \nu_j)^\top \Lambda_{j-1}^{-1} (\phi_j + \nu_j)). \tag{29}$$

Taking the logarithm of both sides of the equation, we utilize the property of logarithms that the logarithm of a product is the sum of the logarithms:

$$\log \det(\Lambda_t) = \log \det(\Lambda_0) + \sum_{j=1}^{t} \log(1 + (\phi_j + \nu_j)^\top \Lambda_{j-1}^{-1} (\phi_j + \nu_j)). \tag{30}$$

We rewrite it as:

$$\log \left[ \frac{\det(\Lambda_t)}{\det(\Lambda_0)} \right] = \sum_{j=1}^{t} \log \left( 1 + (\phi_j + \nu_j)^\top \Lambda_{j-1}^{-1} (\phi_j + \nu_j) \right), \tag{31}$$

and using the property of the logarithm that $\log(1+x) \le x \le 2\log(1+x)$ for $x \in [0,1]$, we have:

$$(\phi_j + \nu_j)^\top \Lambda_{j-1}^{-1} (\phi_j + \nu_j) \le 2 \log \left( 1 + (\phi_j + \nu_j)^\top \Lambda_{j-1}^{-1} (\phi_j + \nu_j) \right). \tag{32}$$

Applying this inequality to the sum, we obtain:

$$\sum_{j=1}^{t} (\phi_j + \nu_j)^\top \Lambda_{j-1}^{-1} (\phi_j + \nu_j) \le 2 \sum_{j=1}^{t} \log \left( 1 + (\phi_j + \nu_j)^\top \Lambda_{j-1}^{-1} (\phi_j + \nu_j) \right). \tag{33}$$

This completes the proof by showing that:

$$\log \left[ \frac{\det(\Lambda_t)}{\det(\Lambda_0)} \right] \le \sum_{j=1}^{t} (\phi_j + \nu_j)^\top \Lambda_{j-1}^{-1} (\phi_j + \nu_j) \le 2 \log \left[ \frac{\det(\Lambda_t)}{\det(\Lambda_0)} \right]. \tag{34}$$

With a similar derivation to the one for Eq. (23), we can conclude the proof of Eq. (26). □

Corollary 1 is indeed a special variant of Lemma 3 with accounting for a case where $\phi_j$ is replaced with $\phi_j + \nu_j$. It is important in RLHF as we estimate the reward model with the BT model over the preference data, which assumes that the preference signal is induced by the reward difference between the preferred response and the dispreferred response (i.e., $\sigma(r(x, y_w) - r(x, y_l))$). Then, under the assumption of linear reward, it further corresponds to the difference in the feature space, i.e., $\phi(x, y_w) - \phi(x, y_l)$.

Now we are ready to prove the main theorem. We restate our main theorem as follows.

**Theorem** (Restatement of Theorem 3). *Assume that for each iteration $1 \le t \le T$, one preference pair sample is collected and added into the dataset in the last iteration $t-1$. Under Assumption 1, if we set $\lambda = 4$ and denote $\iota := \log(1 + (4T)/(d\lambda))$, with probability at least $1 - \delta$, the total regret $\mathrm{Regret}(T)$ after $T$ iterations satisfies*

$$\mathrm{Regret}(T) \le \sqrt{T} \cdot C_1 \cdot \sqrt{\frac{d + \log(1/\delta)}{\gamma^2} + \lambda B^2} \cdot \sqrt{d\iota}$$

*where $C_1$ is an absolute constant.*

*Proof.* For simplicity, we assume that during $T$ iterations in the online RLHF, the dataset which is initialized as an empty set $\mathcal{D}_0 = \emptyset$, is collected according to the following protocol for every $t = 1, ..., T$,

1. Sample one response pair $\{x, y_w, y_l\}$ from $\hat{\pi}_{t-1}$, label the data with the external labeler, and form the dataset $\mathcal{D}_t$ for current iteration $t$ which satisfies $\mathcal{D}_t = \mathcal{D}_{t-1} \cup \{x, y_w, y_l\}$;

2. Compute the output policy $\hat{\pi}_t$ by $\hat{\pi}_t = \arg\max_\pi \hat{J}_\beta(\pi; \mathcal{D}_t)$.

By Theorem 2, we have

$$
\begin{aligned}
\text{Regret}(T) &= \sum_{t=1}^{T} [J_\beta^\star(\pi^\star) - J_\beta^\star(\hat{\pi}_t)] \\
&\leq \sum_{t=1}^{T} 2C \cdot \sqrt{\frac{d + \log(1/\delta)}{\gamma^2 t} + \lambda B^2} \cdot \|\mathbb{E}_{x \sim \rho}[\phi(x, \hat{\pi}_t(x))]\|_{(\Sigma_{\mathcal{D}_t} + \lambda I)^{-1}} \\
&\leq 2C \cdot \sqrt{\frac{d + \log(1/\delta)}{\gamma^2} + \lambda B^2} \cdot \sum_{t=1}^{T} \|\mathbb{E}_{x \sim \rho}[\phi(x, \hat{\pi}_t(x))]\|_{(\Sigma_{\mathcal{D}_t} + \lambda I)^{-1}}.
\end{aligned} \tag{35}
$$

Since we have $\|\phi(x, y)\| \leq 1$ for any $(x, y)$ and $\lambda = 4$, by Corollary 1, we have

$$
\sum_{t=1}^{T} (\mathbb{E}_{x \sim \rho}[\phi(x, \hat{\pi}_t(x))])^\top (\Sigma_{\mathcal{D}_t} + \lambda I)^{-1} (\mathbb{E}_{x \sim \rho}[\phi(x, \hat{\pi}_t(x))]) \leq 2 \log \left[ \frac{\det(\Sigma_{\mathcal{D}_{T+1}} + \lambda I)}{\det(\Sigma_{\mathcal{D}_1} + \lambda I)} \right]
$$

where $\Sigma_{\mathcal{D}_t} = \sum_{i=1}^{t} (\phi(x^{(i)}, y_w^{(i)}) - \phi(x^{(i)}, y_l^{(i)}))(\phi(x^{(i)}, y_w^{(i)}) - \phi(x^{(i)}, y_l^{(i)}))^\top$. Note that the trace of $\Sigma_{\mathcal{D}_t} + \lambda I$ holds

$$
\text{tr}(\Sigma_{\mathcal{D}_t} + \lambda I) = d\lambda + \sum_{i=1}^{t} \|\phi(x^{(i)}, y_w^{(i)}) - \phi(x^{(i)}, y_l^{(i)})\|^2 \leq d\lambda + 4t
$$

where the last inequality results from triangle inequality. Denoting the eigenvalues of $\Sigma_{\mathcal{D}_t} + \lambda I$ by $\{\lambda_i\}_{i=1}^d$, by AM-GM inequality, we thus have

$$
\det(\Sigma_{\mathcal{D}_t} + \lambda I) = \prod_{i=1}^{d} \lambda_i \leq \left( \frac{d\lambda + 4t}{d} \right)^d,
$$

$$
\text{then } \log \det(\Sigma_{\mathcal{D}_t} + \lambda I) \leq d \log(\lambda + \frac{4t}{d}).
$$

At the same time, we have $\log \det(\lambda I) < \log \det(\Sigma_{\mathcal{D}_1} + \lambda I)$, thereby leading to

$$
\sum_{t=1}^{T} \|\mathbb{E}_{x \sim \rho}[\phi(x, \hat{\pi}_t(x))]\|_{(\Sigma_{\mathcal{D}_t} + \lambda I)^{-1}}^2 \leq 2d \log(1 + \frac{4T}{d\lambda})
$$

Introducing $\iota = \log(1 + \frac{4T}{d\lambda})$, by Cauchy-Schwartz inequality, we further have

$$
\begin{aligned}
\sum_{t=1}^{T} \|\mathbb{E}_{x \sim \rho}[\phi(x, \hat{\pi}_t(x))]\|_{(\Sigma_{\mathcal{D}_t} + \lambda I)^{-1}} &\leq \sqrt{T} \cdot \left[ \sum_{t=1}^{T} \|\mathbb{E}_{x \sim \rho}[\phi(x, \hat{\pi}_t(x))]\|_{(\Sigma_{\mathcal{D}_t} + \lambda I)^{-1}}^2 \right]^{1/2} \\
&\leq \sqrt{T} \cdot \sqrt{2d\iota}.
\end{aligned} \tag{36}
$$

Finally, combining Eq. (35) and Eq. (36), we achieve

$$
\text{Regret}(T) \leq \sqrt{T} \cdot C_1 \cdot \sqrt{\frac{d + \log(1/\delta)}{\gamma^2} + \lambda B^2} \cdot \sqrt{d\iota}
$$

where $\iota := \log(1 + \frac{4T}{d\lambda})$ and $C_1$ is an absolute constant. $\qquad\square$

## B  IMPLEMENTATION DETAILS

In optimizing the COPO objective for LLM preference alignment, we adopt Low-Rank Adaptation (LoRA) (Hu et al., 2022) rather than full-parameter training due to limitations in computing resources. In contrast, the results of baseline methods reported in experiments are obtained by

Table 4: Key implementations of the text generation experiments.

| Basic Parameters | |
|---|---|
| Pre-training | Llama-3-8B Instruct & Zephyr-7b-SFT |
| Hardware | 4 x NVIDIA A100 40G |
| Datatype | bfloat16 |
| Fine-tuning strategy | LoRA (Hu et al., 2022) |
| LoRA target module | q-proj & k-proj & v-proj & o-proj & gate-proj & up-proj & down-proj |
| LoRA $r$ | 128 |
| LoRA alpha | 128 |
| LoRA dropout | 0.05 |
| Optimizer | Adamw torch |
| Train epoch | 1 |
| Per device batch-size | 2 |
| Accelerator | Deepspeed Zero3 |
| Learning rate | 5e-7 (1st iter), 3e-7 (2nd iter), 1e-7 (3rd iter) |
| Learning rate scheduler | cosine |
| Learning rate warmup ratio | 0.1 |
| Preference dataset | HuggingFaceH4/ultrafeedback_binarized |
| Reward model | llm-blender/PairRM |
| **CFN (Ours)** | |
| Learning rate | 1e-4 |
| Exploration factor ($\alpha$) | 0.1 (Llama) & 0.01 (Zephyr) |
| Network architecture | 4096 (last hidden-dim for LLM) $\rightarrow 32 \rightarrow 20$ |
| Activation | LeakyReLU & Linear |
| Train epoch | 1 |
| $\lambda$ | 0.01 |

full-parameter tuning, which verifies the effective of our method in low-computation request. The training of our method is conducted on 4xA100-40G GPUs.

The implementation of COPO build on Alignment Handbook implementation (`https://github.com/huggingface/alignment-handbook`), which provides the basic DPO algorithm based on the TRL Repo (`https://github.com/huggingface/trl`). For the implementation of iterative DPO, we adopt the same hyper-parameters as in the SELM implementation (`https://github.com/shenao-zhang/SELM`). Since COPO also adopt online DPO as the backbone, we use the same hyper-parameters as online-DPO except for the CFN network.

COPO mainly contains three stages. (1) **Iterative data collection**. In this stage, we use vLLM `https://github.com/vllm-project/vllm` as the inference server to perform fast sampling of the LLM policy in the previous iteration. In sampling, we set the temperature as 0.0 and the top-p as 1.0. After generating a response for each prompt, we use llm-blender with PairRM reward model (`https://huggingface.co/llm-blender/PairRM`) to rank all three responses (two from the previous dataset and one from sampling). The final dataset contains the best and worst responses as chosen and reject responses, respectively. We also store the response of the LLM policy in the dataset. (2) **CFN training.** We perform CFN training in the prompt-response pair sampled in the dataset using the previous LLM policy. We construct a lightweight CFN network and use it similar to a value head that adheres to an LLM via the 'AutoModelForCausalLMWithValueHead' class. For inference, the CFN takes the last hidden state of the LLM as input and output the prediction of the coin flips. In regression, we adopt the same 'CoinFlipMaker' as in that of online RL (`https://github.com/samlobel/CFN`). We use two independent trainers for CFN and online DPO. (3) **RLHF training.** In this stage, CFN network is keep fixed to provide pseudo-count estimation for the sampled prompt-response pairs, which is used in our exploration objective integrating with online DPO. We summarize the key hyperparameters in Table 4.

## C ADDITIONAL EXPERIMENTS

### C.1 ADVERSARIAL CASES

COPO focuses on improving the exploration ability of the LLM. We conduct an adversarial case that further restricts the data coverage of the initial preference dataset. In this case, the exploratory capacity of LLM becomes particularly important. Specifically, we use a subset of the UltraFeedback dataset with only 20% samples, then we train online DPO, SELM, and COPO for 3 iterations. The performance evaluated on AlpacaEval 2.0 is given in the following table. Based on the results, we find that COPO significantly outperforms other methods where the dataset is quite limited, while other methods have a significant performance drop due to the insufficient exploration ability.

Table 5: Comparison to methods with 20% preference data (LC Win Rate).

| Llama-3-8B-It | DPO (iter1) | DPO (iter2) | DPO (iter3) |
|---|---|---|---|
| 22.92 | 26.60 | 28.01 | 28.16 |
| | SELM (iter1) | SELM (iter2) | SELM (iter3) |
| | 27.81 | 28.79 | 29.25 |
| | COPO (iter1) | COPO (iter2) | COPO (iter3) |
| | 28.32 | 30.14 | 31.80 |

### C.2 COMPARISON TO D2PO

D2PO-related methods also lie on the paradigm of online RLHF, but they mainly focus on how to automatically annotate preference labels for newly generated response pairs of LLMs. Specifically, Self-Rewarding LM (Yuan et al., 2024a) uses the LLM-as-a-Judge ability of LLMs to evaluate response pairs, Judge-augmented SFT (Lee et al., 2024b) trains a pairwise judgment model to output preference label as well as rationale, and Discriminator-Guided DPO (Singhal et al., 2024) trains a discriminative evaluation model to generate annotation for synthetic responses. In contrast, COPO directly adopts an off-the-shelf reward model to rank responses generated by LLMs, which is a 0.4B PairRM of small size in our experiment.

We add D2PO (Singhal et al., 2024) as a baseline by removing the PairRM model and training a discriminator via the Bradley-Terry model. We also change the backbone in D2PO from Llama-2-7B to Llama-3-8B-Instruct. The results on AlpacaEval 2.0 are given as follows. The result shows that the self-trained discriminator achieves competitive performance compared to the online DPO baseline, which uses an off-the-shelf reward model, which signifies the effectiveness of D2PO. However, using a small reward model can be more efficient in practice.

Table 6: Comparison to D2PO with PairRM.

| Method | DPO w/ PairRM Iter1 | DPO w/ PairRM Iter2 | DPO w/ PairRM Iter3 | D2PO Iter1 | D2PO Iter2 | D2PO Iter3 |
|---|---|---|---|---|---|---|
| LC Win Rate | 20.53 | 22.12 | 22.19 | 20.10 | 21.95 | 22.03 |

### C.3 COUNT-BASED EXPLORATION FOR KTO

Exploration is essential for RLHF since the preference data are usually limited in data coverage, which makes DPO develop a biased distribution favoring unseen responses, directly impacting the quality of the learned policy. The proposed exploration objective measures the visitation count of the generated prompt-response pair via a coin-flipping network, which can be combined with various online RLHF algorithms.

In this section, we add a new preference optimization objective in addition to DPO for experiments, i.e., KTO (Ethayarajh et al., 2024a). KTO maximizes the utility of generations, rather than just the likelihood of preferences. It works effectively with binary feedback, which is more abundant and easier to collect than the preference data that the DPO requires. We adopt the implementation of the KTO loss function from TRL [1] and use online iterations for KTO similar to online DPO. We find

---

[1] https://huggingface.co/docs/trl/kto_trainer

that the count-based objective in COPO can also be combined with the online KTO algorithms to further enhance its performance.

Table 7: The result comparison of online KTO and count-based KTO.

|  | KTO (iter1) | KTO (iter2) | KTO (iter3) | KTO+ COPO (iter1) | KTO+COPO (iter2) | KTO+COPO (iter3) |
|---|---|---|---|---|---|---|
| LC Win Rate | 33.19 | 35.40 | 35.90 | 35.32 | 36.84 | 37.10 |

## C.4 ABLATION OF REWARD MODEL

Due to the fact that online RLHF methods require an additional reward module to give preference labels compared to the offline method, we adopt a small-size reward model to ensure that the additional requirement is minimal. Specifically, We follow the SELM baseline to use a small-sized PairRM (0.4B) model to rank responses generated by LLM and contain the best and worst responses according to the reward model. As we mentioned in our paper, using a powerful reward model can further improve performance. As a result, we add additional experiments that use a powerful 8B reward model from RewardBench (Lambert et al., 2024), named *RLHFlow/ArmoRM-Llama3-8B-v0.1* (Dong et al., 2024) for comparison. The learned models evaluated by AlpacaEval 2.0 are given in the following Table. The result shows that both COPO and SELM have improved in performance, with COPO showing a more significant improvement. The reason is that the data coverage of generated responses in COPO is broader than SELM since we adopt count-based exploration in alignment. Then an accurate reward model is required for preference labeling in such a large data coverage.

Table 8: The ablation of more powerful reward model with *ArmoRM-Llama3-8B*.

|  | SELM (w/ PairRM-0.4B) | SELM (w/ ArmoRM-8B) | COPO (w/ PairRM-0.4B) | COPO (w/ ArmoRM-8B) |
|---|---|---|---|---|
| LC Win Rate | 34.67 | 39.21 | 35.54 | 41.78 |

## C.5 DISCUSSION OF BEST-OF-N AND PPO

The Best-of-N policy is a method for aligning samples from LLMs to human preferences. It involves drawing $N$ samples, ranking them, and returning the best. For best-of-N, we consider two types of methods to integrate this strategy into the RLHF process. (1) Using best-of-N to improve the quality of preference data in *offline RLHF*. Specifically, we use prompts from the UltraFeedback dataset and regenerate the chosen and rejected response pairs $(y_w, y_l)$ with the LLM. For each prompt $x$, we generate $N$ responses using the SFT model with a low sampling temperature (e.g., 0.8 in the experiment). We then use PairRM to score these responses, selecting the highest-scoring one as $y_w$ and the lowest-scoring one as $y_l$. We denote this method as *DPO-best*. (2) Using best-of-N as an exploration method in *online RLHF*. In each iteration of online DPO, we denote the current LLM policy as $\pi_1$ and the best-of-N variant as $\pi_2$. In this way, the $\pi_2$ policy increases the margins between $\pi_1$ and provides more exploration. We denote this variant by *online-DPO-best*.

The comparison evaluated in AlpacaEval 2.0 is given in the following table, where we use $n = 5$. We find that the best-of-N strategy significantly improves the performance of the offline DPO method, which improves the coverage of the offline dataset. Meanwhile, the best-of-N strategy also enhances online DPO also improves the performance of online DPO. We note that although best-of-N provides another efficient exploration strategy, it is more computationally expensive and still underperforms our method, which signifies that the proposed count-based objective is an efficient and stronger exploration approach. We do not apply the best-of-N strategy in evaluation since all methods follow the same evaluation pipeline in a standard benchmark, including AlpacaEval 2.0, MT-Bench, and LLM leaderboard.

As for PPO, (1) we note that reproducing the successful RLHF results with PPO is challenging as it requires extensive efforts and resources that the open-source communities usually cannot afford, which has been discussed in previous works (Ivison et al., 2024). Specifically, the PPO algorithm requires loading multiple LLMs simultaneously, including the actor (policy), critic (value network), reward model, and reference model (for KL estimation), which places significant pressure on GPU

Table 9: The combination of DPO/online-DPO with Best-of-N

| - | DPO | DPO-best | online DPO | online DPO best | ours |
|---|---|---|---|---|---|
| LC Win Rate | 22.5 | 26.0 | 33.17 | 34.12 | 35.54 |

memory that exceeds resources in our group. (2) Meanwhile, the performance of PPO usually heavily relies on the quality of the reward model. Due to the narrow distribution coverage of the preference dataset, reward misspecification often occurs, which makes the reward model assign a high value to out-of-distribution (OOD) samples and has the potential to be exploited during the RL process. In contrast, using a better reward model (Lambert et al., 2024) trained on a larger dataset will significantly boost the performance, while the comparison to the DPO-based method becomes unfair since it leverages knowledge beyond the fixed preference dataset.

## D    DISCUSSION OF OTHER EXPLORATION METHODS

The use of count-based bonus $1/\sqrt{N(s)}$ was originally proposed in tabular cases to encourage exploration in RL, where the count $N(s)$ of each state $s \in \mathcal{S}$ can be calculated accurately since the total number of states are finite. However, for environments with high-dimensional state space, we cannot obtain the exact count for states since the state space is infinite, thus a pseudo-counting mechanism is required to estimate the $\hat{N}(s)$. In online RL exploration literature (), several methods have been proposed to estimate the pseudo-counts, including neural density model, hash code, random-network distillation (RND), and CFN. However, we find only CFN is suitable for LLM to estimate the pseudo count of prompt-response pairs for the following reasons.

(1) For the neural density models (Bellemare et al., 2016), we have $\hat{N}(s) = (e^{\mathrm{PG_n(x)}} - 1)^{-1}$, where $\mathrm{PG_n(x)}$ is the prediction gain that measures density changes after using the specific state to update the density model. This calculation process is hard to implement in LLM because training a density model for prompt-response pairs is highly expensive (Ostrovski et al., 2017), and it is also difficult to measure density changes by using a single prompt-response pair to update the LLM. (2) For hash code (Tang et al., 2017), it requires mapping prompt-response pairs to a hash table, and the method assigns exploration bonuses based on the frequency of state visits. This approach is also hard to implement since it requires training an autoencoder (AE) that takes a prompt-response pair as inputs and outputs the hash code in the latent space, then the AE is required to reconstruct the prompt-response pair. Meanwhile, it can suffer from hash collisions, especially for the response space of LLMs, potentially leading to inaccurate exploration objectives. (3) RND (Burda et al., 2018) is a popular exploration method in RL, determined by the output difference between a randomly initialized network and a trained network's prediction of the state. Despite its simplicity and empirical success, RND's exploration bonus is challenging to interpret, as it is based on an unnormalized distance within a neural network's latent space. Additionally, there are no rigorous theoretical results to prove the RND reward is exactly the pseudocount of the state.

Compared to previous works, CFN (Lobel et al., 2023) represents a significant advance in exploration methods using the Rademacher distribution to estimate pseudo counts. This approach does not rely on density estimation but instead uses the averaging of samples from the Rademacher distribution to derive state visitation counts. CFN's key advantages include its theoretical grounding, which allows it to provide exploration bonuses equivalent to count-based objectives, and its simplicity and ease of training. Unlike other methods, CFN does not impose restrictions on the function approximator or training procedure, offering flexibility in model architecture selection. In our work, we set the CFN as a lightweight network with several fully connected layers on top of a pre-trained hidden layer. The empirical result in online RL also demonstrates CFN's effectiveness, particularly in complex environments, and its ability to generalize well to novel states.

