# OpenReview forum: "Online Preference Alignment for Language Models via Count-based Exploration"
_ICLR.cc/2025/Conference — ICLR 2025 Spotlight_

### Official Review · Reviewer_pkVP · 2024-10-30

**Soundness:** 2
**Presentation:** 2
**Contribution:** 2
**Rating:** 6
**Confidence:** 3

**Summary:**

This paper introduces an online RLHF approach via the Count-based Online Preference Optimization (COPO) algorithm, which integrates count-based exploration and direct preference optimization to improve performance. Experiments on Zephyr and Llama-3 models demonstrate COPO’s effectiveness in boosting instruction-following capabilities and general academic benchmarks.

**Strengths:**

The paper rigorously establishes the theoretical underpinnings of COPO, providing a regret-bound analysis that demonstrates the efficiency of the UCB-based exploration strategy.

**Weaknesses:**

It seems there’s no notable improvement over SELM for the Llama-3-8B Instruct model in Table 1, especially when considering the potential variance from using GPT-4 as the evaluator. And can authors provide more baselines, for example comparison with Best-of-N and PPO?

**Questions:**

Please see the weakness.

---

> ### Author Response · Authors · 2024-11-21
>
> We appreciate your valuable comments and would like to address each of your concerns.
>
> **1. It seems there’s no notable improvement over SELM for the Llama-3-8B Instruct model in Table 1, especially when considering the potential variance from using GPT-4 as the evaluator.**
>
> Thanks for your comment. We remark that the first term in the COPO objective is the original DPO objective, and the second term is the count-based exploration term. According to the experiment results, our method obtains much better results than online DPO due to the use of exploration terms. Specifically, COPO increases the LC win rate of AlpacaEval 2.0 from 22.19 to 27.21 (*18.8\% improvement*) for Zephyr-7B, and increases the LC win rate from 33.17 to 35.54 (*7.1\% improvement*) for Llama3-8B-It, which we believe is a significant improvement in the instruction-following tasks.
>
> Meanwhile, (1) we remark that AlpacaEval [1] is an LLM-based automatic evaluation that is widely used in the RLHF community. As shown in their document [2], AlpacaEval 2.0 demonstrates a high correlation with human annotations, which is crucial for establishing its credibility as an evaluation tool. The Spearman correlation of AlpacaEval 2.0 with ChatBot Arena is an impressive 0.98, indicating a very strong positive correlation. This means that the rankings produced by AlpacaEval 2.0 closely match those that would be produced by human evaluators. (2) These responses are compared to reference responses by the provided GPT-4-based auto-annotators. GPT-4 is employed to compare model responses against a reference model's outputs, determining which is preferred more frequently. The improvements made in AlpacaEval 2.0 include enhancing the auto-annotator to be more cost-effective and accurate, further enhancing the reliability of the evaluation process. (3) Consistency is very important in evaluation frameworks to ensure that the results are stable and replicable. AlpacaEval 2.0 addresses this by using a standardized process for comparing model outputs to reference responses. The evaluation set, AlpacaFarm, while diverse, is designed to test models across a broad range of simple instructions, ensuring a consistent benchmark for model performance.
>
> We run the other two seeds of the COPO experiment for tuning Llama3-8B-Instruct, and evaluate the performance via AlpacaEval 2.0. The results are given in the following Table. We have made the code open-source at https://github.com/review-anon/COPO to ensure the reproducibility of the results.
>
> | | Seed1 | Seed2 |  Seed3 |
> |--|--|--|--|
> | LC Win Rate | 35.54 | 35.78 | 35.51 |
>
>
> [1] https://tatsu-lab.github.io/alpaca_eval/
>
> [2] https://github.com/tatsu-lab/alpaca_eval
>
> **2. And can authors provide more baselines, for example comparison with Best-of-N and PPO?**
>
> Thanks for the comment. We discuss Best-of-N and PPO as follows. Meanwhile, we add online KTO as a baseline by following other reviewer's comments.
>
> As for best-of-N, we consider two types of methods to integrate this strategy into the RLHF process. (1)  Using best-of-N to improve the quality of preference data in *offline RLHF*. Specifically, we use prompts from the UltraFeedback dataset and regenerate the chosen and rejected response pairs $(y_w, y_l)$ with the LLM. For each prompt $x$, we generate $N$ responses using the SFT model with a low sampling temperature (e.g., 0.8 in the experiment). We then use PairRM to score these responses, selecting the highest-scoring one as $y_w$ and the lowest-scoring one as $y_l$. We denote this method as *'DPO-best'.* (2) Using best-of-N as an exploration method in *online RLHF*. In each iteration of online DPO, we denote the current LLM policy as $\pi_1$ and the best-of-N variant as $\pi_2$. In this way, the $\pi_2$ policy enlarges the margins between $\pi_1$ and provides more exploration, as verified in [1]. We denote this variant as *'online-DPO-best'*. The comparison evaluated in AlpacaEval 2.0 is given in the following table, where we use $n=5$. We find the best-of-N strategy significantly improves the performance of offline DPO method, which improves the coverage of the offline dataset. Meanwhile, the best-of-N strategy also enhances online DPO also improves the performance of online DPO. We note that although best-of-N provides another efficient exploration strategy, it is more computationally expensive and still underperforms our method, which signifies that the proposed count-based objective is an efficient and stronger exploration approach. (3) We do not apply the best-of-N strategy in evaluation since all methods follow the same evaluation pipeline in a standard benchmark, including AlpacaEval 2.0, MT-Bench, and LLM leaderboard.
>
> | - | DPO | DPO-best | online DPO |  online DPO best | ours |
> | - | -------- | --------  | --------  | -------- | -------- |
> | LC Win Rate | 22.5  | 26.0  | 33.17  | 34.12  | 35.54  |

---

> ### Author Response · Authors · 2024-11-21
>
> As for PPO, (1) we would like to note that reproducing the successful RLHF results with PPO is challenging for the open-source community as it requires extensive efforts and resources that the open-source communities usually cannot afford, which has been discussed in previous works [2]. Specifically, the PPO algorithm requires loading multiple LLMs simultaneously, including the actor (policy), critic (value network), reward model, and reference model (for KL estimation), which places significant pressure on GPU memory that exceeds resources in our group. (2) Meanwhile, the performance of PPO usually heavily relies on the quality of the reward model. Due to the narrow distribution coverage of the preference dataset, reward misspecification often occurs, which makes the reward model assign a high value to out-of-distribution (OOD) samples and has the potential to be exploited during the RL process. In contrast, using a better reward model [3] trained on a larger dataset will significantly boost the performance, while the comparison to the DPO-based method becomes *unfair* since it leverages knowledge beyond the fixed preference dataset. We will add more discussion about this point.
>
> Meanwhile, we remark that COPO armed with a count-based exploration term can even outperform much larger LLMs, such as Yi-34B-Chat and Llama3-70B-Instruct in LC win rate, according to Table 1 in the paper.
>
> **References**
>
> [1] Dong H, Xiong W, Pang B, et al. Rlhf workflow: From reward modeling to online rlhf. arXiv preprint arXiv:2405.07863, 2024.
>
> [2] Ivison H, Wang Y, Liu J, et al. Unpacking DPO and PPO: Disentangling Best Practices for Learning from Preference Feedback. NeurIPS 2024
>
> [3] Lambert N, Pyatkin V, Morrison J, et al. Rewardbench: Evaluating reward models for language modeling. arXiv preprint arXiv:2403.13787, 2024.

---

> > ### Author Response · Authors · 2024-11-26
> >
> > Dear Reviewer:
> >
> > We wanted to express our gratitude for your insightful feedback during the review process of our paper. We hope we have resolved all the concerns and showed the improved quality of our paper. Please do not hesitate to contact us if there are other clarifications we can offer.
> >
> > Best,
> >
> > The authors.

---

### Official Review · Reviewer_j1HG · 2024-11-01

**Soundness:** 3
**Presentation:** 2
**Contribution:** 3
**Rating:** 8
**Confidence:** 4

**Summary:**

This paper identifies a central problem in current online RLHF methods, that is, efficiently exploring the prompt-response space in each iteration. To address this, the paper designs a novel RLHF algorithm named Count-based Online Preference Optimization (COPO), which re-formulates the original DPO objective to balance space exploration and preference optimization. The authors adopt a simple coin-flip counting network approach to support COPO's pseudo-count, achieving promising experimental results.

**Strengths:**

1. Solid theoretical analysis. This work is built upon a solid theoretical foundation, starting from a relatively reasonable linear reward assumption and ultimately analyzing the regret upper bound of the optimization objective, providing a more credible motivation.

2. The authors demonstrate good insights into the existing problems in online RLHF, specifically the focus on preference optimization while lacking effective space exploration. Around this issue, they propose an algorithm suitable for LLMs with vast token sequence spaces.

3. The proposed COPO algorithm shows notable improvements after multiple iterations.

**Weaknesses:**

1. The proposed COPO algorithm is only implemented based on the DPO optimization objective, without verification of its adaptability to improve other online RLHF algorithms.

2. The visit count item in the COPO algorithm relies solely on a pseudo-count mechanism implemented by a coin flipping network (CFN) with accompanying random labels, which seems too simplistic. More convincing visit count algorithms need to be explored.

3. As mentioned by the authors in the experimental section, a better reward model could improve COPO's performance. However, this point wasn't explored in the experiments. In fact, the choice of different scoring RMs has a significant impact on the experiment and should be given more attention.

**Questions:**

1. What would be the experimental results if COPO's main ideas were applied to other online RLHF algorithms?

2. Are there other alternative or explorable counting algorithms?

3. How would choosing a stronger Reward model (or Judge model) affect algorithm performance, and can this be demonstrated with specific experimental results?

---

> ### Author Response · Authors · 2024-11-21
>
> Thank you for your positive feedback and recognition of our work. We appreciate your valuable comments and would like to address each of your concerns.
>
> **1.  The proposed COPO algorithm is only implemented based on the DPO optimization objective, without verification of its adaptability to improve other online RLHF algorithms.**
>
> Thanks for the comments.  The contribution of our paper lies in proposing a count-based objective in online RLHF to enhance the exploration ability of the LLM policy. Exploration is essential for RLHF since the preference data are usually limited in data coverage, which makes DPO develop a biased distribution favoring unseen responses, directly impacting the quality of the learned policy. The proposed exploration objective measures the visitation count of the generated prompt-response pair via a coin-flipping network, which can be combined with various online RLHF algorithms.
>
> Following the reviewer's suggestion, we add a new preference optimization objective in addition to DPO for experiments, i.e., KTO [1]. KTO maximizes the utility of generations rather than just the likelihood of preferences. It operates effectively with binary feedback, which is more abundant and easier to collect than the preference data that the DPO requires. We adopt the implementation of the KTO loss function from TRL (https://huggingface.co/docs/trl/kto_trainer) and use online iterations for KTO similar to online DPO.  We find that the count-based objective in COPO can also be combined with the online KTO algorithms to further enhance its performance.
>
> |  | KTO (iter1) |  KTO (iter2) | KTO (iter3) | KTO + COPO (iter1) | KTO + COPO (iter2) | KTO + COPO (iter3) |
> |--|--|--|--|--|--|--|
> | LC Win Rate | 33.19 | 35.40 | 35.90 | 35.32 | 36.84 | 37.10 |
>
>
> **2.  The visit count item in the COPO algorithm relies solely on a pseudo-count mechanism implemented by a coin flipping network (CFN) with accompanying random labels, which seems too simplistic. More convincing visit count algorithms need to be explored.**
>
>
> Thanks for the question. The use of count-based bonus $1/\sqrt{N(s)}$ was originally proposed in tabular cases to encourage exploration in reinforcement learning (RL), where the count $N(s)$ of each state $s\in \mathcal{S}$ can be calculated accurately since the total number of states are finite. However, for environments with high-dimensional state space, we cannot obtain the exact count for states since the state space is infinite, thus a pseudo-counting mechanism is required to estimate the $\hat{N}(s)$. In online RL exploration literature, several methods have been proposed to estimate the pseudo-counts, including neural density model, hash code, random-network distillation (RND), and CFN. However, we find only CFN is suitable for LLM to estimate the pseudo count of prompt-response pairs for the following reasons.
>
> - For the neural density model [2], we have $\hat{N}(s)=(e^{\rm PG_n(x)}-1)^{-1}$, where $\rm PG_n(x)$ is the prediction gain that measures the density changes after using the specific state to update the density model. Such a calculating process is hard to implement in LLM because training a density model for prompt-response pairs is highly expensive [3], and it is also hard to measure the density changes by using a single prompt-response pair to update the LLM.
> - For hash code [4], it requires mapping prompt-response pairs to a hash table, and the method assigns exploration bonuses based on the infrequency of state visits. This approach is also hard to implement since it requires training an autoencoder (AE) that takes a prompt-response pair as inputs and outputs the hash code in the latent space, then the AE is required to reconstruct the prompt-response pair from this code, which can be highly expensive to train. Meanwhile, it can suffer from hash collisions, especially for the response space of LLMs, potentially leading to inaccurate exploration objectives.
> - RND is a popular exploration method in RL [5], determined by the output difference between a randomly initialized network and a trained network's prediction of the state. Despite its simplicity and empirical success, RND's exploration bonus is challenging to interpret, as it is based on an unnormalized distance within a neural network's latent space. Additionally, there are no rigorous theoretical results to prove the RND reward is exactly the pseudo-count of the state.

---

> ### Author Response · Authors · 2024-11-21
>
> Compared to previous works, CFN [6] represents a significant advancement in exploration methods by leveraging the Rademacher distribution to estimate pseudo counts. This approach does not rely on density estimation but instead uses the averaging of samples from the Rademacher distribution to derive state visitation counts. CFN's key advantages include its theoretical grounding, which allows it to provide exploration bonuses equivalent to count-based objectives, and its simplicity and ease of training. Unlike other methods, CFN does not impose restrictions on the function approximator or training procedure, offering flexibility in model architecture selection. In our work, we set the CFN as a lightweight network with several fully connected layers on top of a pre-trained hidden layer. The empirical result in online RL also demonstrates CFN's effectiveness, particularly in complex environments, and its ability to generalize well to novel states.
>
> **References**
>
> [1] Ethayarajh K, Xu W, Muennighoff N, et al. Kto: Model alignment as prospect theoretic optimization. ICML 2024
>
> [2]  Bellemare M, Srinivasan S, Ostrovski G, et al. Unifying count-based exploration and intrinsic motivation. NeurIPS 2016
>
> [3] Ostrovski G, Bellemare M G, Oord A, et al. Count-based exploration with neural density models. ICML 2017
>
> [4] Tang H, Houthooft R, Foote D, et al. #Exploration: A study of count-based exploration for deep reinforcement learning. NeurIPS 2017
>
> [5] Burda Y, Edwards H, Storkey A, et al. Exploration by random network distillation. ICLR 2019
>
> [6] Lobel S, Bagaria A, Konidaris G. Flipping coins to estimate pseudocounts for exploration in reinforcement learning. ICML 2023
>
> **3.  As mentioned by the authors in the experimental section, a better reward model could improve COPO's performance. However, this point wasn't explored in the experiments. In fact, the choice of different scoring RMs has a significant impact on the experiment and should be given more attention.**
>
> Thanks for the question. Due to the fact that online RLHF methods require an additional reward module to give preference labels compared to the offline method, we adopt a small-size reward model to ensure the additional requirement is minimal. Specifically, We follow the SELM baseline to use a small-sized PairRM (0.4B) model to rank responses generated by LLM and contain the best and worst responses according to the reward model. As we mentioned in our paper, using a powerful reward model can further improve performance. As a result, we add additional experiments that use a powerful 8B reward model from Reward Bench [1], named *RLHFlow/ArmoRM-Llama3-8B-v0.1* [2] for comparison. The learned models evaluated by AlpacaEval 2.0 are given in the following Table. The result shows that both COPO and SELM have improved in performance, with COPO showing a more significant improvement. The reason is that the data coverage of generated responses in COPO is broader than SELM since we adopt count-based exploration in alignment. Then an accurate reward model is required for preference labeling in such a large data coverage.
>
> |  | SELM (w/ PairRM-0.4B) | SELM (w/ ArmoRM-8B) | COPO  (w/ PairRM-0.4B) | COPO (w/ ArmoRM-8B) |
> |--|--|--|--|--|
> | LC Win Rate | 34.67  |  39.21 | 35.54  | 41.78  |
>
>
> [1] https://huggingface.co/spaces/allenai/reward-bench
>
> [2] https://huggingface.co/RLHFlow/ArmoRM-Llama3-8B-v0.1

---

> > ### Comment · Reviewer_j1HG · 2024-11-24
> >
> > Thank you to the authors for the response, which has largely addressed my concerns. I believe there are still better exploration methods compared to CFN, and I hope this issue can be resolved in the future. For now, I will raise the score.

---

> > > ### Author Response · Authors · 2024-11-26
> > > **Thanks**
> > >
> > > Thank you for supporting our work! Indeed, with the advancements in online RL and RLHF algorithms, we anticipate the development of improved exploration algorithms for LLMs in the future. Here, we outline two potential approaches to address this challenge:
> > > - Bayesian LLM: Traditional RL employs Bayesian neural networks [1,2] to estimate the posterior distribution of the value function, which implicitly restores pseudo-counting for input states. However, the development of Bayesian LLMs has not been thoroughly explored within the community.
> > > - Bootstrapped LLM: Bootstrapped or ensemble neural networks [3,4] have demonstrated their ability to quantify state uncertainty in online RL exploration. In the context of LLMs, using different prompts, inference parameters, dropout, or hypernetworks may implicitly achieve a similar effect, but further justification is needed.
> > >
> > > [1] Houthooft R, Chen X, Duan Y, et al. Vime: Variational information maximizing exploration. NeurIPS 2016
> > >
> > > [2] Li Z, Li Y, Zhang Y, et al. HyperDQN: A randomized exploration method for deep reinforcement learning. ICML 2021
> > >
> > > [3] Osband I, Blundell C, Pritzel A, et al. Deep exploration via bootstrapped DQN. NeurIPS 2016
> > >
> > > [4] Bai C, Wang L, Han L, et al. Principled exploration via optimistic bootstrapping and backward induction. ICML 2021

---

### Official Review · Reviewer_4QFJ · 2024-11-04

**Soundness:** 3
**Presentation:** 3
**Contribution:** 3
**Rating:** 8
**Confidence:** 3

**Summary:**

This paper introduces the COPO algorithm for better exploration of online RLHF formulation. The authors provide a theoretical background of the proposed method and explain the practical implementation of it. To be specific, it assumes that the reward function could be expressed as a dot product between a parameter vector and a feature vector. Based on it, the model incorporates a USB term for exploration. Also, they employ a lightweight Config Flipping Network (CFN) to obtain pseudo-visit counts for the large state space of language generation. Experimental results show the efficacy of the proposed COPO algorithm. It performs better than the other exploration-aware online RLHF baseline, SELM, in several benchmarks. When they adjust the coefficient for the exploration term, it supports the importance of the exploration term in their experiments.

**Strengths:**

It tackles the important problem innate in RLHF-based alignment learning, i.e., exploration, with formal theoretical motivation and practical algorithm.

**Weaknesses:**

* Their explanation and selection of baseline methods are not enough. Especially, I think other iterative/online methods should have been
considered in addition to the SELM. There are many variants to implement online DPO. For example, [1], [2], and [3].

* The improvements seem marginal despite the complexity of the proposed method. The performance gaps among the baselines could be changed with different random seeds, especially for the LLM-as-a-Judge tasks.

* It would be good if more benefits of the exploration were suggested. For example, it might be more robust in some adversarial cases.

[1] Self-Rewarding Language Models (Yuan et al., 2024)

[2] Aligning Large Language Models by On-Policy Self-Judgment (Lee et al., 2024)

[3] D2PO: Discriminator-Guided DPO with Response Evaluation Models (Singhal et al., 2024)

**Questions:**

* How do we measure/believe the CFN works properly? (before testing the model on the final benchmarks.)
  * (Declaimer) I do not have any idea about the original CFN work. However, the model (CFN module) seems too small to represent the hidden state of the modern LLM.
  * If we can improve the performance of the module, will the final alignment performances be also improved?

---

> ### Author Response · Authors · 2024-11-21
>
> **1. Their explanation and selection of baseline methods are not enough.... There are many variants to implement online DPO. For example, [1], [2] and [3].**
>
> Thanks for your comments. In COPO, we adopt a novel count-based exploration term combined with DPO to address the fundamental problem in online RLHF. Specifically, since the initial preference dataset can have limited state coverage, performing systematic exploration in online RLHF is important to explore the space of token sequences and collect informative experiences that could benefit LLM alignment.
>
> The suggested methods [1-3] also lie on the paradigm of online RLHF, but they mainly focus on how to automatically annotate preference labels for newly generated response pairs of LLMs. Specifically, Self-Rewarding LM [1] uses the LLM-as-a-Judge ability of LLMs to evaluate response pairs, Judge-augmented SFT [2] trains a pairwise judgment model to output preference label as well as the rationale, and Discriminator-Guided DPO [3] trains a discriminative evaluation model to generate annotation for synthetic responses. In contrast, COPO directly adopts an off-the-shelf reward model to rank responses generated by LLMs, which is a small-size 0.4B PairRM in our experiment.
>
> As a result, COPO and suggested references [1-3] contribute online RLHF from different perspectives, i.e., exploration and automatic ranking, respectively. On the one hand, COPO can be combined with other ranking/judgment/reward models that can give preference annotations. On the other hand, the count-based exploration term proposed in COPO can enhance other online RLHF algorithms by increasing the coverage of preference data. We choose SELM as a baseline since it serves as a state-of-the-art online RLHF algorithm, and more importantly, it also aims to enhance the exploration ability of online RLHF. In contrast, works in [1-3] contribute to online RLHF from an orthogonal perspective.
>
> We add D2PO [3] as a baseline by removing the PairRM model and training a discriminator via the Bradley-Terry model. We also change the backbone in D2PO from Llama-2-7B to Llama-3-8B-Instruct. The results on AlpacaEval 2.0 are given as follows. The result shows that the self-trained discriminator obtains competitive performance compared to the online DPO baseline in our paper, which uses an off-the-shelf reward model, which signifies the effectiveness of D2PO. However, using a small reward model can be more efficient in practice.
>
> | - | DPO w/ PairRM Iter1| DPO w/ PairRM Iter2 | DPO w/ PairRM Iter3 | D2PO Iter1 | D2PO Iter2 | D2PO Iter3 |
> | - | -------- | --------  | --------  | -------- | -------- | -------- |
> | LC Win Rate | 20.53  | 22.12  | 22.19  | 20.10  | 21.95  | 22.03  |
>
> [1] Self-Rewarding Language Models (Yuan et al., 2024)
>
> [2] Aligning Large Language Models by On-Policy Self-Judgment (Lee et al., 2024)
>
> [3] D2PO: Discriminator-Guided DPO with Response Evaluation Models (Singhal et al., 2024)
>
> ~
>
> **2. The improvements seem marginal despite the complexity of the proposed method. The performance gaps among the baselines could be changed with different random seeds, especially for the LLM-as-a-Judge tasks.**
>
> Thanks for your comment. We remark that the optimization objective of COPO is $\max J_{copo}(\pi_\phi) = -L_{DPO}(\pi_\phi) + \alpha \mathbb{E}[1/(\sqrt{N_{D_t}(x,y)+\lambda})]$, as shown in Eq. (16). The first term is the original DPO objective, and the second term is proposed to encourage count-based exploration. According to the experiment results, our method obtains much better results than online DPO due to the introduction of the exploration term. Specifically, COPO increases the LC win rate of AlpacaEval 2.0 from 22.19 to 27.21 (*18.8\% improvement*) for Zephyr-7B, and increases the LC win rate from 33.17 to 35.54 (*7.1\% improvement*) for Llama3-8B-It, which we believe is a significant improvement in the instruction-following tasks.
>
> Our method is trained using the UltraFeedback 60K dataset, which comprises human feedback and preferences specifically about instruction-following or task execution. Since the dataset focuses on how well the model follows instructions or completes tasks as intended by humans, the LLMs with RLHF can improve the instruction-following ability significantly on similar benchmarks (e.g., AlpacaEval 2.0 and MT-Bench). However, such an alignment process might not necessarily align well with the requirements of academic benchmarks, which often require abstract reasoning, complex inference, or extensive factual knowledge that may not be sufficiently enhanced by feedback focused on instruction following. In experiments, we find our method outperforms DPO slightly, which signifies the alignment also helps increase the model's capabilities in some academic areas, while it does not automatically mean the model will perform better in solving complex math problems (e.g., GSM8K).

---

> ### Author Response · Authors · 2024-11-21
>
> We add additional runs in experiments and find the evaluation results of COPO among multi-seeds are quite consistent. Because COPO can explore uncertain prompt-response pairs in each iteration, it results in a broader coverage of the whole preference dataset $\cup_t D_t$. Then the DPO objective is easier to find the best policy in such a space with wide state coverage.
>
>
> **3. It would be good if more benefits of the exploration were suggested. For example, it might be more robust in some adversarial cases.**
>
> Thanks for the suggestion. Since we focus on enhancing the exploration ability of the LLM, we add an adversarial case that further restricts the data coverage of the initial preference dataset. In this case, the exploratory ability of LLM becomes particularly important. Specifically, we use a subset of the UltraFeedback dataset with only 20% samples, then we train online DPO, SELM, and COPO for 3 iterations. The performance evaluated on AlpacaEval 2.0 is given in the following table. According to the results, we find that COPO significantly outperforms other methods where the dataset is quite limited, while other methods have a significant performance drop due to the insufficient exploration ability.
>
> | w/ 20% data | Llama-3-8B-It |DPO (iter1) | DPO (iter2) | DPO (iter3) | SELM (iter1) | SELM (iter2) | SELM (iter3) | COPO (iter1) | COPO (iter2) | COPO (iter3) |
> |--|--|--|--|--|--|--|--|--|--|--|
> | LC Win Rate | 22.92 | 26.60 | 28.01 | 28.16 | 27.81 | 28.79 | 29.25 | 28.32 | 30.14 | 31.80 |
>
>
> **4. How do we measure/believe the CFN works properly? (before testing the model on the final benchmarks.**
>
> In online RL, compared to previous exploration methods that require estimating the density model or state auto-encoder, CFN represents a significant advancement in exploration methods by leveraging the Rademacher distribution to estimate pseudo counts. This approach does not rely on density estimation but instead uses the averaging of samples from the Rademacher distribution to derive state visitation counts. CFN's key advantages include its theoretical grounding, which allows it to provide exploration bonuses equivalent to count-based objectives, and its simplicity and ease of training. Unlike other methods, CFN does not impose restrictions on the function approximator or training procedure, offering flexibility in model architecture selection. In our work, we set the CFN as a lightweight network with several fully connected layers on top of a pre-trained hidden layer.
>
> In our training process, the CFN network is trained on the generated prompt-response dataset in each iteration. Since the target label of each output dimension in CFN is generated from {-1,1}, the estimation error for pre-dimension will be converged to 1 finally. As a result, we follow this principle to design the network, which ensures the pre-dimensional CFN loss can converge to 1 in a few training epochs. As in our experiments, we find a three-layer network is sufficient to make CFN network coverage by leveraging the feature vector extracted by the LLM. As a result, the performance cannot be improved by increasing the model capacity of CFN since it will not minimize the loss of CFN further. The empirical result in online RL also demonstrates CFN's effectiveness, particularly in complex environments, and its ability to generalize well to novel states.

---

### Author Response · Authors · 2024-12-02
**General Response**

Dear reviewers and AC,

We sincerely appreciate your valuable time and effort spent reviewing our manuscript. We propose Count-based Online Preference Optimization (COPO) for LLM alignment that leverages coin-flip counting to encourage exploration in online RLHF with theoretical guarantees. As reviewers highlighted, our method has solid theoretical motivation (4QFJ, j1HG, pkVP), good insights and practical algorithm (j1HG, 4QFJ), and notable improvements (j1HG, pkVP). We appreciate your constructive feedback on our manuscript. In response to the comments, we have carefully revised and enhanced the manuscript as follows:
- we add experiments on D2PO, an automatic ranking method compared to the off-the-shelf PairRM reward model.
- we add experiments on an adversarial case that restricts the data coverage of the initial preference dataset to show the benefit of our exploration objective.
- we add experiments on online KTO with count-based exploration, which serves as a new preference optimization objective in addition to the DPO-based method.
- we add experiments on a more powerful reward model, *ArmoRM-Llama3-8B*, which shows that a better reward model can enhance the online RLHF algorithms.
- we add experiments on both offline and online methods to integrate the Best-of-N strategy into the RLHF process.

Meanwhile, we give more clarifications of the algorithmic details and experiments as follows:
- we add an explanation of the measurement of CFN in the training process and the estimation error of CFN after coverage.
- we add an explanation of the marginal improvement on some specific tasks in academic benchmarks.
- we add an explanation of other exploration algorithms to show why CFN is more suitable for online RLHF algorithms.
- we add an explanation of the consistency results evaluated by AlpacaEval 2.0.

Meanwhile, we wish to highlight that Reviewer pkVP with negative evaluations did not give any feedback on our response in the rebuttal period. The major concern of Reviewers pkVP is the potential variance from using GPT-4 as the evaluator and the lack of baselines of Best-of-N and PPO. In the rebuttal period, we add a detailed analysis of the consistency of AlpacaEval 2.0 which has a diverse evaluation set and a good alignment to human evaluators. Meanwhile, we add Best-of-N as a baseline that combines offline RLHF and online RLHF methods. We also give explanations on the PPO-based method. We believe our response has successfully addressed the concerns raised by the reviewers. We kindly request that the AC dedicate more attention to the evaluation of our submission.

In the revised manuscript, these updates are temporarily highlighted in brown for your convenience to check. We sincerely believe that these updates may help us better deliver the benefits of the proposed COPO to the ICLR community.

Thank you very much,

Authors.

---

### Meta-Review · Area_Chair_qsLw · 2024-12-16

**Metareview:**

This paper addresses the practical challenge of improving online preference alignment. The authors begin by identifying limitations in existing methods and then provide a theoretical foundation for their proposed approach. Building on this theoretical framework, they introduce a novel alignment method based on a simple count-based approach.  The experimental results demonstrate clear and consistent performance gains.

A key strength of this work is the strong theoretical grounding for the proposed method, which effectively addresses the fundamental exploration-exploitation trade-off in reinforcement learning systems.  Furthermore, the authors translate this theoretical foundation into a novel yet simple alignment method that proves effective across various datasets.

While the paper presents compelling results, there is potential for further improvement. Exploring more sophisticated reward models could enhance the method's performance.  Additionally, expanding the empirical evaluation could provide a more comprehensive understanding of the approach's capabilities and limitations.

**Additional Comments On Reviewer Discussion:**

The authors effectively addressed the reviewers' concerns during the rebuttal period, resolving nearly all of the issues raised. The remaining comments primarily consist of suggestions for further improvement, which can be incorporated into the camera-ready version or explored in future work.

---

### Decision · Program_Chairs · 2025-01-22

Accept (Spotlight)